evolution, ecology, genomics

conservation genomics, demographic inference, effective population size, Bayesian skyline

**Authors for correspondence:**

Andrew J. Helmstetter
e-mail: andrew.j.helmstetter@gmail.com
Alexander S. T. Papadopulos
e-mail: a.papadopulos@bangor.ac.uk

# The demographic history of Madagascan micro-endemics: have rare species always been rare?

Andrew J. Helmstetter[1,2], Stuart Cable[1,3], Franck Rakotonasolo[3], Romer Rabarijaona[3], Mijoro Rakotoarinivo[4], Wolf L. Eiserhardt[1,5], William J. Baker[1] and Alexander S. T. Papadopulos[1,6]

[1]Royal Botanic Gardens, Kew, Richmond, Surrey TW9 3AE, UK
[2]Institut de Recherche pour le Développement (IRD), UMR-DIADE, 911 Avenue Agropolis, BP 64501, Montpellier 34394, France
[3]Kew Madagascar Conservation Centre, Lot II J 131 B Ambodivoanjo, Ivandry, Antananarivo 101, Madagascar
[4]Mention Biologie et Ecologie Végétales, Faculté des Sciences, Université d'Antananarivo, Antananarivo BP 906101, Madagascar
[5]Department of Biology, Aarhus University, Aarhus, Denmark
[6]Molecular Ecology and Evolution Bangor, Environment Centre Wales, School of Natural Sciences, Bangor University, Bangor LL57 2UW, UK

AJH, 0000-0003-3761-4981; ASTP, 0000-0001-6589-754X

Extinction has increased as human activities impact ecosystems, yet relatively few species have conservation assessments. Novel approaches are needed to highlight threatened species that are currently data-deficient. Many Madagascan plant species have extremely narrow ranges, but this may not have always been the case—it is unclear how the island's diverse flora evolved. To assess this, we generated restriction-site associated DNA sequence data for 10 Madagascan plant species, estimated effective population size ($N_e$) for each species and compared this to census ($N_c$) sizes. In each case, $N_e$ was an order of magnitude larger than $N_c$—signifying rapid, recent population decline. We then estimated species' demographic history, tracking changes in $N_e$ over time. We show that it is possible to predict extinction risk, particularly in the most threatened species. Furthermore, simulations showed that our approach has the power to detect population decline during the Anthropocene. Our analyses reveal that Madagascar's micro-endemics were not always rare, having experienced a rapid decline in their recent history. This casts further uncertainty over the processes that generated Madagascar's exceptional biodiversity. Our approach targets data-deficient species in need of conservation assessment, particularly in regions where human modification of the environment has been rapid.

## 1. Introduction

The existence of a mass extinction during modern times has become well accepted [1]. According to the International Union for Conservation of Nature (IUCN) red list, approximately 6% of described species have been assessed and 28% are threatened by extinction [2]. With fewer than 135 k species assessed, we have poor knowledge of the extinction risk in many taxa, including those that are vital to human wellbeing. For example, only 54 k plant species have IUCN assessments (approx. 13% of all plants), but the loss of plant diversity is occurring at a pace higher than background rates [3] and is expected to have a greater effect on humans than the loss of diversity in any other group [4]. Therefore, it is critical that we improve the accuracy and efficiency of plant species assessments.

Despite the rigour with which assessments are made, it is possible that current methods substantially underestimate the extent to which species may be at risk of extinction. For example, population decline within the last 100 years is a fundamental component of how IUCN red listing is performed [5,6], but recent estimates suggest that 30% of all vertebrate species are in decline and as much as 55% of declining bird species are not classified as threatened [1]. Furthermore, substantial population decline in many species may have pre-dated historical records or may not have been observed directly at all. As a result, it is unclear whether species that are rare today have experienced population decline or simply have a low carrying capacity (e.g. after becoming highly specialized on spatially restricted habitats).

Genomic data can provide an important perspective on the conservation status of the species yet are typically underused [7]. By inferring demographic patterns with sequence data we can make comparisons with risk levels inferred from other data types and provide novel information. For instance, the effective population size ($N_e$) of a species is normally substantially smaller than the census population size, such as in humans, where $N_e$ is in the order of 10 000 [8]. When populations are experiencing rapid decline it is possible for $N_e$ to substantially exceed census population size, as has been observed in the critically endangered black rhinoceros [9]. In species that have not yet been assessed, genomic data can be included at the outset to achieve more robust conservation assessments. For instance, genetic signals of decline could identify species that are more at risk of genomic erosion, which may be hidden from other assessment methods. Without taking into account unobserved declines in population sizes we may be drastically underestimating the number of threatened species and the severity of the extinction threat they face. The unique and highly endemic flora of Madagascar provides an opportunity to investigate the link between historical demographic change and extinction risk.

Levels of endemism on Madagascar are some of the highest worldwide; all of the island's amphibians and terrestrial mammals, 83% of vascular plant species and 86% of macro-invertebrates are found nowhere else [10–12]. A striking feature of the Madagascan flora and fauna is the abundance of micro-endemic species that have small ranges and are relatively rare [10,13,14]. For example, more than half of the island's endemic palms are known to be extremely rare—many are found in just a single population or with fewer than 100 individuals in the wild [15]. The processes that gave rise to the incredible diversity and micro-endemism of Madagascar are hotly debated [10,13,16–20], but proposed models put little emphasis on the role human beings have played in shaping the distribution of species. Humans arrived on Madagascar recently (estimates range from 10 000 to 1000 years ago [21–24]) and preceded the rapid extinction of the island's megafauna [25]. Although there are alternative explanations for megafaunal population decline [26,27], increasing human activities (including hunting, slash and burn agriculture, illegal logging and introduction of invasive species) are likely to have played a major role [28]. Extinction of large lemuriforms is also thought to have had a negative impact on the dispersal and fitness of large-seeded plants [29]. Additionally, deforestation has advanced so rapidly that only 10% of the island's forest remains [30]. This combination of recent human influence and abundant micro-endemism invites the question: have Madagascar's micro-endemics always been rare, or is this pattern driven by exceptional population decline across the island's flora and fauna?

We assessed population size changes in 10 Madagascan plant species using genome-wide genetic markers. Our sample included a range of endemic species from the humid and littoral forests of eastern Madagascar (figure 1; electronic supplementary material, table S1). These forests have decreased in area by up to a third since the 1970s and are severely under threat [32]. We put special focus on four members of the diverse palm (Arecaceae) flora of Madagascar. This group is well studied and more than 90% of the 204 Madagascan palms have been placed in one of the threatened categories following red list assessment—an indication that these all have declining or small populations. In this study, our aims were to (i) investigate whether Madagascan plants have suffered population declines in the recent past and (ii) determine whether inference of past demographic changes from genetic data could be a useful predictor of extinction risk.

## 2. Material and methods

### (a) Sample collection

Forty-three individuals across 10 species and seven families were sampled (electronic supplementary material, table S1). Leaf tissue was collected from wild individuals and desiccated using silica gel to preserve DNA. Approximately 20 mg of dried tissue was used for DNA extractions. The majority of samples were collected in Andasibe, Madagascar, and palm species were collected across the island (figure 1). Location data are detailed in electronic supplementary material, table S1. All samples were collected and exported with prior informed consent of Madagascan authorities under all necessary permits, including CITES permits where appropriate. Permit numbers can be found in electronic supplementary material, table S1.

### (b) Library preparation

Genomic DNA was extracted using hexadecyltrimethylammonium bromide (CTAB) mini-extraction protocol [33], purified using spin columns from the Qiagen DNeasy Plant Mini Kit and then eluted in 60 µl water. Double-digest restriction-site associated DNA (ddRAD) libraries were constructed following the protocol of Peterson et al. [34]. Briefly, 1 µg of high-molecular weight DNA was digested using two rare-cutting enzymes; EcoRI-HF (NEB) and SphI-HF (NEB). Barcoded flex adapters [34] were ligated to 400 ng digested DNA for multiplexing and samples were pooled. Size selection was carried out with a Pippin Prep (Sage Biosciences) with a tight window of 468–546 bp. Eight PCR reactions per library were conducted to minimize the effect of PCR bias.

### (c) Sequencing and genotyping

Libraries were sequenced on three Illumina MiSeq runs, using $2 \times 75$ bp (V3), $2 \times 250$ bp (V2) and $2 \times 300$ bp (V3) kits. Read pairs were concatenated to produce a single sequence of 147 bp ($2 \times 75$ bp), 380 bp ($2 \times 250$ bp) or 598 bp ($2 \times 300$ bp) after barcode removal and trimming of low-quality bases (all loci belonging to a species were of the same length). We performed a de novo analysis in the software Stacks (v. 1.40) [35]. Reads were demultiplexed using process_radtags, dropping low-quality reads under the default options. Reads were aligned and SNPs

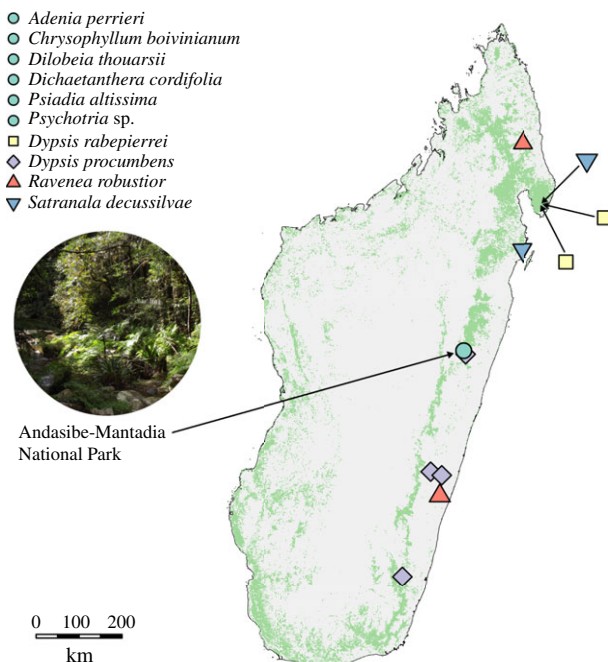

- ⊙ *Adenia perrieri*
- ⊙ *Chrysophyllum boivinianum*
- ⊙ *Dilobeia thouarsii*
- ⊙ *Dichaetanthera cordifolia*
- ⊙ *Psiadia altissima*
- ⊙ *Psychotria* sp.
- ▢ *Dypsis rabepierrei*
- ◇ *Dypsis procumbens*
- △ *Ravenea robustior*
- ▽ *Satranala decussilvae*

Andasibe-Mantadia
National Park

0   100   200
km

**Figure 1.** A map of collection sites for the 10 species used in this study. Collection sites are shape and colour coded for each locale and species. The six non-palm species (green circles) were collected from a single location, Andasibe-Mantadia National Park, shown in the inset photo on the left. Light green colouring indicates forest cover in 2017, taken from Vieilledent *et al.* [31]. Inset photo was modified from original by Smiley.toerist and shared under CC-BY-SA-4.0 (https://creativecommons.org/licenses/by-sa/4.0/). (Online version in colour.)

were detected using *ustacks* with the minimum depth of coverage required to create a stack set to '−m 5' and the maximum distance allowed between stacks set to '−M 4'. Diversity measures can be inflated by factors such as the merging of paralogous sequences so we also employed the deleveraging algorithm (−d) to avoid over-merging tags. A separate catalogue was built for each species (*cstacks*; default options), to which loci were matched from each individual of a species (*sstacks*; default options). *Populations* were used to filter and output data by only retaining those loci that were present in all individuals within a species (options '−p 1' and '−r 1'). Given that we were working with a small number of individuals we retained as much sequence data as possible—no minor allele frequency filter was used and we kept monomorphic loci. Information on ploidy in our species was not readily available so we examined haplotype distributions and found no evidence of polyploidization in our species.

### (d) Inference of $N_e$

For each species, the fasta format output from the Stacks pipeline containing all aligned loci was used to estimate effective population size. We first estimated the population scaled mutation rate ($\Theta = 4N_e\mu$) using a Bayesian method implemented in the R package *thetamateR*, which permits rate variation between loci and makes use of all ddRAD loci per species. The estimated $\Theta$ value for each species was converted to $N_e$ assuming the average mutation rate in angiosperms ($5.35 \times 10^{-9}$ sites year$^{-1}$ [36]) and a generation time of either 10 or 50 years. Estimates of $N_e$ converted from $\Theta$ are dependent on the mutation rate ($\mu$) used. Ideally, we would use $\mu$ estimated specifically for the species of interest or their close relatives but such information is not yet available for the vast majority of species, so a general rate for angiosperms is applied here.

### (e) Extended Bayesian skyline plots

To infer changes in population size through time, we used the extended Bayesian skyline plot (EBSP) [37] approach in BEAST v. 2.4.2 [38], following the methodology outlined by Trucchi *et al.* [39], for each species separately. A custom R script [39] was used to select only those loci with high numbers of SNPs (three or more per locus) to maximize the amount of information per marker to increase the accuracy and reliability of our inference. Datasets used for EBSP differed from *thetamateR*, for which all ddRAD loci were used. Due to computational restraints of the EBSP approach, 50 loci were selected randomly to use in the EBSP analysis. One haplotype was chosen at random for each individual at heterozygous loci to minimize bias in haplotype frequencies and reduce bias caused by undiscovered heterozygote samples [39]. A separate substitution model was assigned for each SNP category (e.g. all loci with three SNPs represents one SNP category) with kappa linked between models, trees were unlinked across loci and a single strict clock was assigned to all loci. Operators were modified to increase mixing efficiency [39]. Each MCMC ran for 500 000 000 generations, sampling every 50 000 and the level of stationarity (*Ravenea robustior* was run for twice this length). Stationarity was verified using Tracer v. 1.6 [40]. Multiple replicate runs were conducted to ensure the same stationarity point was reached and runs combined to increase ESS values to suitable levels (ESS greater than 100 for all important parameters). *EBSPanalyser* in BEAST was applied to perform a linear reconstruction with data from combined runs. Six of our 10 species had an additional, previously unused set of 50 highly variable loci, so we repeated EBSP analyses and results compared across datasets to assess consistency.

### (f) Simulations of anthropogenic population decline

We used fastsimcoal2 (v. 2.6) [41] to simulate sequence data for genetic bottlenecks at 100, 250 and 500 generations ago (equivalent to 1000, 2500, 5000 years). We simulated four strengths of bottleneck: 90%, 50% and 10% reductions and a 10% increase in effective population size with ancestral population size set to 25 000. We generated 40 000 DNA sequences matching the length of the corresponding empirical RAD tags (147 bp, 380 bp) and selected loci with the same distribution of SNPs in the empirical *Dypsis procumbens* dataset (i.e. each SNP category contained the same number of tags). We then used these loci as input for our pipeline for EBSP, replacing the output of *populations*. We repeated simulations for each decline severity at least three times to assess consistency in demographic patterns across different simulations.

## 3. Results and discussion

### (a) Effective population size greatly exceeds census size in palms

We used four palm species (all of which have decreasing or small populations) as our 'training' dataset—their red list assessments provide prior expectations for the demographic patterns we estimate from genetic data. Two of these have been assessed as near threatened with decreasing population trends—*R. robustior* [42] is estimated to have fewer than 1000 mature individuals in the wild, while *D. procumbens* is more common but restricted to 32 locations and is exploited for timber [43]. *Dypsis rabepierrei* has yet to be redlisted but is expected to be considered critically endangered with no more than 20 individuals known from a single location [44].

*Satranala decussilvae* is endangered with fewer than 200 mature trees in the wild [45].

If the studied palm species have always been rare, we expected that $N_e$ would be equal to, or less than, the census population sizes. Conversely, if rare species were more abundant prior to human influences in Madagascar $N_e$ would be substantially higher than contemporary census sizes. We genotyped 2–5 individuals of each species at 24–54 k loci (see electronic supplementary material, table S2 and methods for details) and used these data to estimate $N_e$. In each of the palm species with census data, $N_e$ substantially exceeds the current species-wide census population sizes (figure 2; electronic supplementary material, table S3), a strong indication that these rare species have not always been rare. It is possible that the downstream inferences of $N_e$ may have been overestimated due to the merging of paralogous sequences inflating diversity. However, the Stacks pipeline limits over-merging of loci through filtering high coverage loci and the number of paralogous sites [46] present will only be inflated if the choice of $M$ (maximum distance between stacks) is too high, here we chose an intermediate value [47]. Heterozygosity does not appear to have been inflated (electronic supplementary material, table S2 and figure S1), although we have few samples in each dataset to assess this fully. Likewise, when working with rare species it may be difficult to accurately estimate census sizes for this comparison, but given the sizable disparity between $N_e$ and $N_c$ in all palm species (electronic supplementary material, table S3) it is unlikely that unaccounted individuals, or paralogy, would alter the inferred pattern.

## (b) Demographic trends predict extinction risk

To provide further evidence that these palm populations have experienced declines, we used a subset of loci to investigate the demographic history through time with a coalescent-based approach, EBSP. Given that our $N_e$ estimates suggest our four palm species have rapidly decreasing populations, we expected that species population sizes over time would decrease sharply towards the present. The endangered palm species have very small census sizes, so we expected that any decline would be more pronounced in these species. If so, this approach could provide a useful indicator of extinction risk.

For each species, we assessed whether we could reject a hypothesis of constant size through time by assessing the 'sum(indicators.alltrees)' statistic, which is analogous to the number of population size change events. When 95% highest posterior density (HPD) values do not include zero this indicates that $N_e$ has not been constant through time, and we were able to reject constant size for all palm species (electronic supplementary material, table S2). We did not find evidence of a recent decline in the near-threatened palm species (figure 2), each with a single population size change event, corresponding to early population expansion. It is possible that these lower risk species are more abundant than thought. However, such methods struggle to estimate $N_e$ in the very recent past due to a paucity of coalescent events and maintenance of variation in diminished populations, so declines very close to the present may not be inferred. Given the large difference between $N_e$ and census sizes (electronic supplementary material, table S3), there could have been a very rapid, recent decline that could not be detected with EBSP.

On the other hand, the endangered palm species have experienced very steep population declines towards the present, as much as a 90% reduction in $N_e$ (figure 2; electronic supplementary material, table S4), and multiple demographic events in their history. These results are expected for an endangered species, so we suggest that the approach used here is a useful tool for predicting extreme levels of extinction risk where population decline may have been more severe. We note that contraction estimates were based on median values, and in some species (e.g. *D. rabepierrei* or *R. robustior*) HPD intervals were wide towards the present, indicating some uncertainty in their demographic history in the very recent past, so caution should be used when interpreting decline proportions.

## (c) Simulations reveal declines at anthropogenic time scales

To validate whether our approach has the power to detect anthropogenic population decline, we simulated sequence datasets at a range of population declines of varying severity that took place 250 generations (2500 years ago). We then inferred $N_e$ over time using EBSP, as for empirical data. We found that the number of demographic events ('sum(indicators. alltrees)') was greater than one for all declines simulated at 90% severity. The lower 95% HPD of these simulations did not overlap with 0, indicating that population size was not constant over time (electronic supplementary material, table S5). As the severity of the decline decreased (i.e. when $N_e$ was more stable), we were unable to reject constant size more often. Constant sizes were consistently rejected at all severities for 147 bp simulations, even at smaller simulated changes in population size. This suggests that the 'sum(indicators.alltrees)' is not as informative when using shorter sequences. The EBSP-estimated rates of decline were as expected given the original simulated percentage (figure 3; electronic supplementary material, figure S2). For example, trends estimated using EBSP for 90% contraction exhibited the steepest decline. Estimates of the extent of decline were highly correlated with simulated proportions at 380 bp (figure 3; Spearman's rank correlation, $S = 335.59$, $\rho = 0.706$, $p > 0.001$), but less so at 147 bp (electronic supplementary material, figure S2; $S = 655.08$, $\rho = 0.507$, $p = 0.022$). We note that expansions appear to be very difficult to detect in the recent past, as even when expansions are simulated declines can be inferred (figure 3a), particularly when they occurred more recently (electronic supplementary material, figure S4a) and when smaller RADtag sizes are used for inference (electronic supplementary material, figure S2a).

## (d) How do generation times and mutation rates affect inferences?

It remains difficult to accurately estimate a generation time in rare rainforest species, particularly due to lack of relevant life-history data. Factors such as overlapping generations and seed setting rates in the crowded rain forest habitat complicate the calculation of generation times. We used 10-year generation times for palms, which is consistent with other studies in date and *Howea* palms [48,49]. This also seemed an appropriate estimate for those species where we lack specific information on generation times, based on evidence from plants with similar habits and from the same families

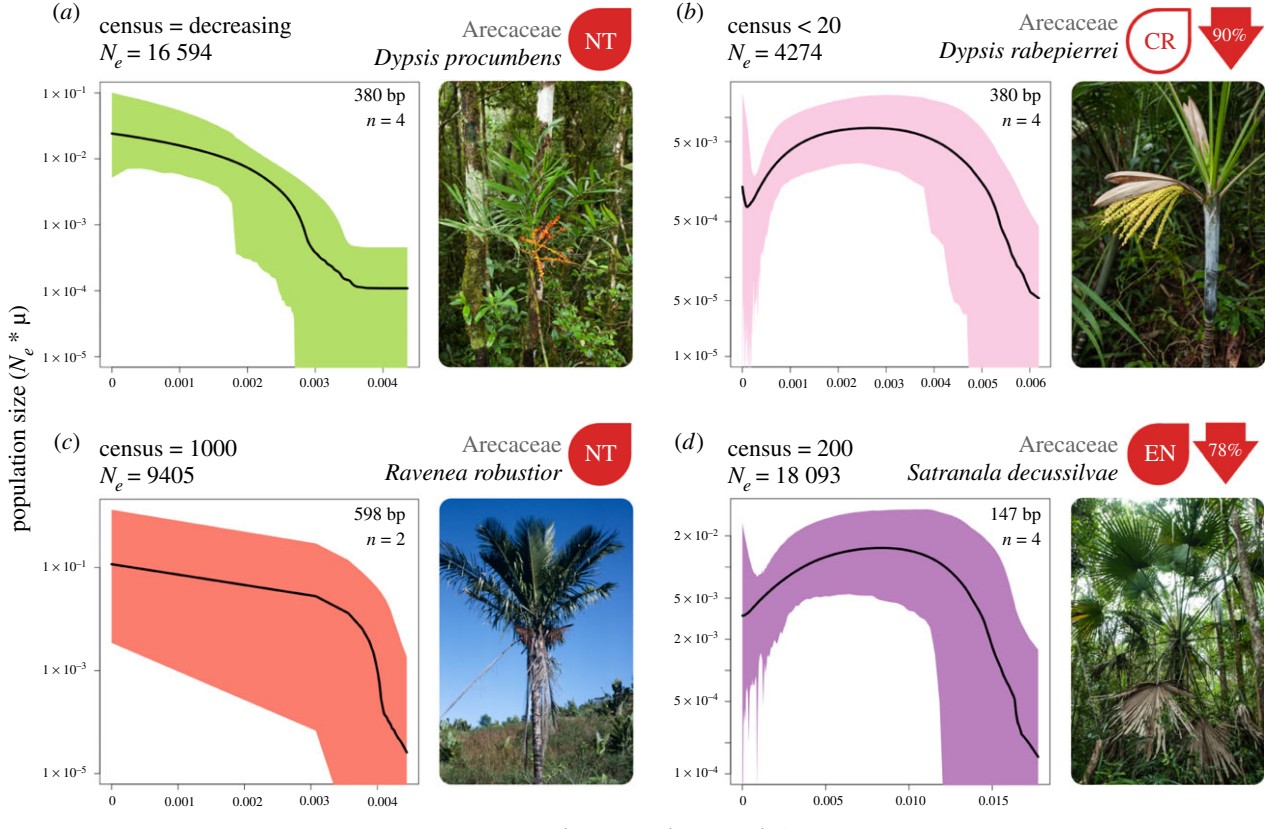

**Figure 2.** Effective population size (*y*-axis) through time (*x*-axis) for four species of palm that have conservation assessments. The present day is on the left side of each plot. Black lines represent the median population size and shaded polygons show the 95% highest posterior density (HPD). To the right of each plot is a picture of the study species and the current IUCN red list assessment or likely assessment category (NT, near threatened; EN, endangered; CR, critically endangered). Values in red arrows indicate the percentage decline from maximum median population size to the minimum median population size inferred after the maximum. The total read length in base pairs (bp) and the number of individuals for each species's dataset is indicated in the top right of each box. The increase in the median value of population size for *D. rabepierrei* in the very recent past is likely to be the result of a lack of coalescent events, which is reflected by extremely wide 95% HPD values during this period, and should not be interpreted as population size increase. Photo of *R. robustior* by Henk Beentje. All other photos by W.J.B. (Online version in colour.)

as our study species (e.g. [50,51]). A known mutation rate is also involved in the conversion of $\Theta$ to $N_e$, and this is also difficult to estimate without extensive genomic resources in the target species, making it a potential source of error. However, for the confidence intervals of $N_e$ estimates for an endangered species like *S. decussilvae* to overlap with census sizes would require a decrease by a factor of 88 and it is extremely unlikely that misspecified mutation rates and generation times would have such a strong effect. In species with larger census sizes like *Dichaetanthera cordifolia*, a smaller, but still unlikely, 25-fold change would be required.

To limit the effect of misspecified generation times we also estimated $N_e$ with a generation time of 50 years and found that previously observed patterns held (electronic supplementary material, table S3). We performed additional simulations, placing demographic events at 100 (electronic supplementary material, figures S3 and S4) and 500 (electronic supplementary material, figures S5 and S6) generations to test how changes in time scale affected our ability to infer population size trends with EBSP. Indeed, we had difficulty in distinguishing between decline severities over short time scales (100 generations), as expected, while over longer periods (500 generations) our results were similar to those inferred at 250 generations (figure 3, electronic supplementary material, figures S2, S5 and S6). This indicates that there is a lower age limit on inferences of demographic changes with EBSP and ddRADseq data.

We observed a reduction in population size close to the very recent past (less than 0.005 substitutions site$^{-1}$) in almost all runs regardless of simulated decline severity. In part, this is likely to be due to a lack of signal caused by limited data and low numbers of coalescent events near the present. However, this was not observed in the empirical EBSP results for *D. procumbens* (figure 2*a*) and maybe an artefact of simulating constant population size (rather than growth) followed by a very recent shift in $N_e$. Nevertheless, these results show that our approach has the power to detect population decline in the Anthropocene and identify the severity of the bottleneck, particularly when using longer sequence lengths. While we are not able to provide exact timings for demographic events, results from our simulations (figure 3) present a comparable timeframe to our empirical results (figures 2 and 4), indicating that events probably occurred less than 500 generations ago.

### (e) Diverse demographic histories across species from seven plant families

To investigate the prevalence of population decline in less well-studied Madagascan species, we repeated the above analyses for the remaining six species in our dataset (figure 4)—only one of which has a known census population size. Among this group, just four species have been red list assessed and three are categorized as of least concern:

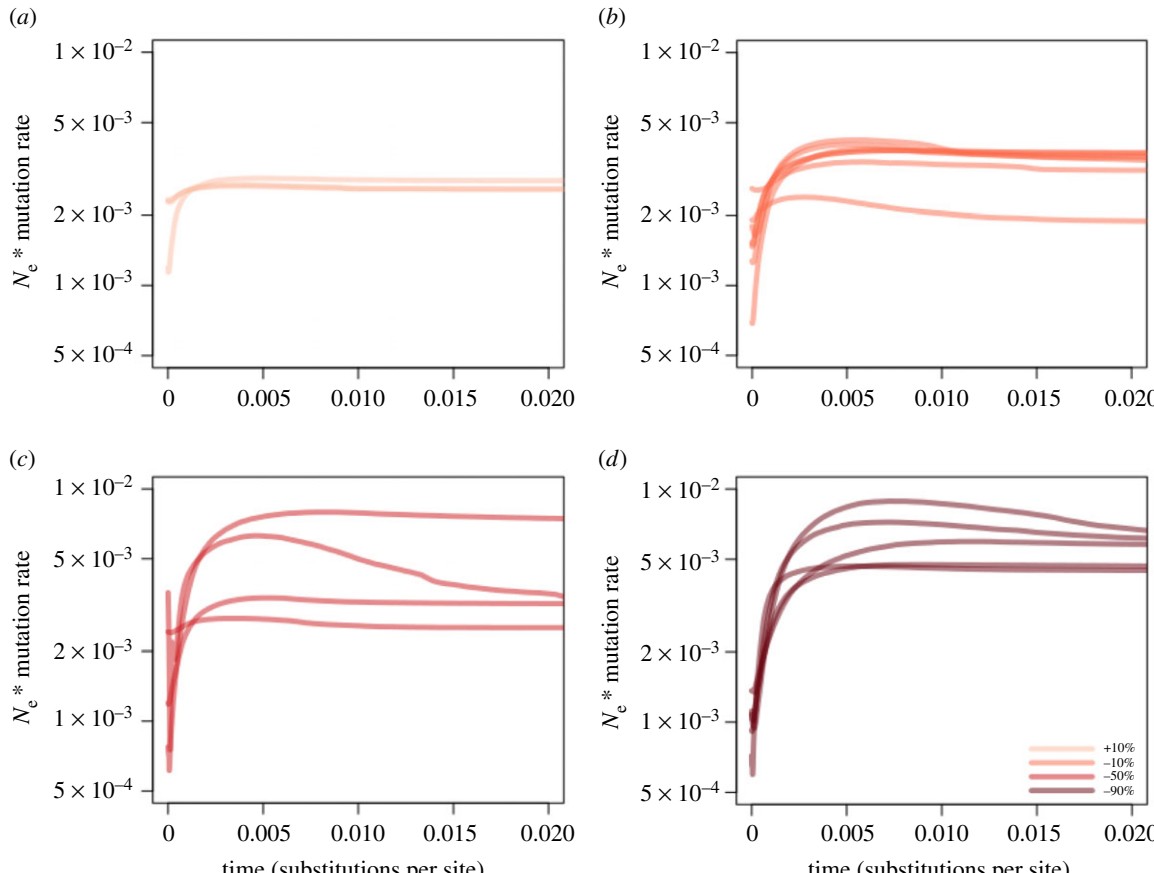

**Figure 3.** Simulated effective population size through time for one expansion ((a) +10%) and three different levels of population decline ((b) −10%, (c) −50%, (d) −90%) at 250 generations (2500 years with a generation time of 10 years) before present. We performed two sets of simulations: using two different sequences length of 380 bp (shown here) and 147 bp (shown in electronic supplementary material, figure S2). Simulated data was then used in the same EBSP pipeline as previously used for empirical data. The different lines in each plot represent different simulation runs. We ran simulations until we had at least three separate EBSP runs that converged per level of decline, resulting in (a) 3, (b) 7, (c) 4 and (d) 5 simulations shown here. The x-axis shows time measured in substitutions per site (present on the left side of the graph) and the y-axis shows $N_e$ scaled by mutation rate. Shades of red deepen according to increasing severity of simulated decline, as shown in the legend of panel (d). (Online version in colour.)

*Psiadia altissima* and *D. cordifolia* have stable populations (*P. altissima* is a slash-and-burn successor), while *Dilobeia thouarsii* is decreasing due to agriculture, harvesting for timber and traditional medicine [52,53]. *Chrysophyllum boivinianum* is also exploited for timber and malarial medicine and is suspected to be declining rapidly [54]. Little is known about the extent of exploitation and biology of the undescribed species of *Psychotria*. In these six species, $N_e$ ranged from 7254 in *Psychotria* sp. to 25 702 in *D. cordifolia* (figure 4). Although census sizes are not available for these populations, it seems unlikely that any of these species would have always been rare, indeed *P. altisimma* is relatively common [55]. However, the EBSP analyses are more revealing. Two species showed no evidence of recent decline; *D. cordifolia*, as expected given its conservation status, and *Adenia perrieri*. The inferred history of the latter is more surprising given its vulnerable status, but could be explained by a very recent decline that was undetected in the EBSP and caused by contemporary processes such as slash-and-burn agriculture [56]. *Psiadia altissima* showed some evidence of decline (↓43%) but we could not reject constant size for this species. The remaining three species showed clear decreases in population size towards the present, which is consistent with the suspected exploitation of two of these, *C. boivinianum* (↓83%) and *D. thouarsii* (↓82%). Madagascar is a centre on endemism for *Psychotria* (↓68%), which are

often small understory trees and shrubs with limited ranges so these species may be particularly susceptible to deforestation. We were able to repeat EBSP analyses with alternate marker sets in six species (electronic supplementary material, figure S7) and found results that were consistent with initial demographic trends (figures 2 and 4), suggesting that our results are robust to potential marker choice bias. However, as our simulations have shown, there is a greater degree of uncertainty for the analyses where short sequences were used (e.g. *P. altissima* and *C. boivinianum*). Finally, we compared percentage declines in our empirical data to our simulations (electronic supplementary material, table S4) and found that empirical declines fell within the range of simulated declines (10–90% decline) in all cases. This suggests our simulations represent the range of patterns inferred from empirical data.

## (f) Potential effects of population structure on demographic inferences

Results from skyline plot methods where samples are taken from only one of several genetic clusters can be confounded by effects of population structure, exhibiting false evidence of population decline [57] when population sizes have, in fact, been constant. Despite this uncertainty, $N_e$ versus $N_c$ comparison adds corroborating evidence to support declines

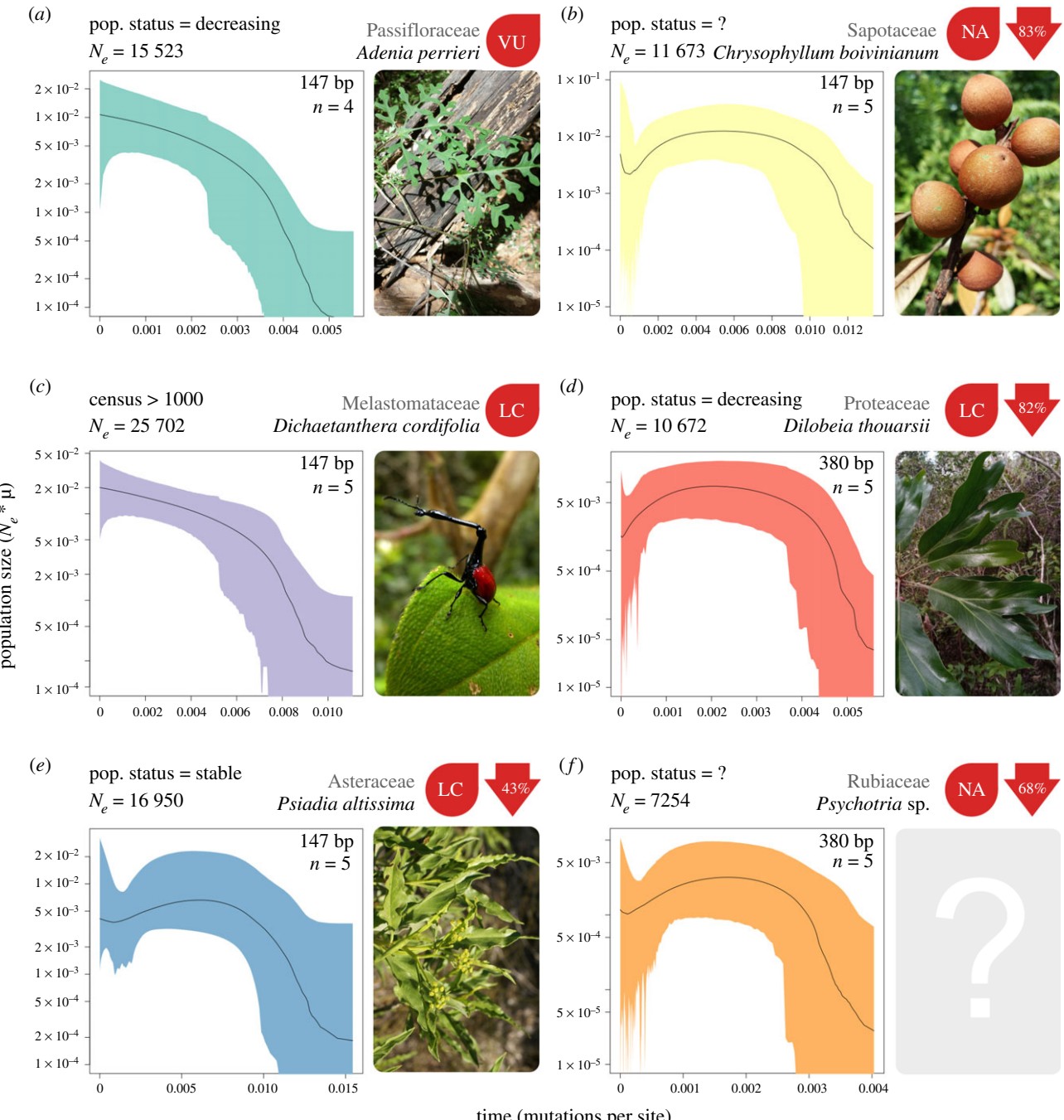

**Figure 4.** Effective population size through time for six species from six different plant families. The present day is on the left side of each plot. Black lines represent the median population size and shaded polygons show the 95% highest posterior density (HPD). To the right of each plot is a picture of the study species and the current IUCN red list assessment, where available (NA, not assessed; LC, least concern; VU, vulnerable). Values in red arrows indicate the percentage decline from maximum median population size to the minimum median population size inferred after the maximum. The total read length in base pairs (bp) and the number of individuals for each species's dataset is indicated in the top right of each box. As described in figure 1, rapid increases towards the present in *C. boivinianum* and *Psychotria* sp. are likely to be due to lack of signal. Photos of *A. perrieri*, *C. boivinianum* and *D. thouarsii* by FR, Solofo Eric Rakotoarisoa and Rahaingoson Fabien respectively and shared under the CC BY-NC 4.0 license (https://creativecommons.org/licenses/by-nc/4.0/). Photo of *D. cordifolia* by Frank Vassen and shared under the CC BY 2.0 license (https://creativecommons.org/licenses/by/2.0/deed.en). Photo of *P. altissima* by B.navez shared under the CC BY-SA 3.0 license (https://creativecommons.org/licenses/by-sa/3.0/). Small modifications have been made to each photo. No photo was available for the undescribed species *Psychotria* sp. (Online version in colour.)

in locally sampled species like those observed in *D. rabepierrei*, *C. boivinianum* and *D. thouarsii*. Furthermore, population structure may have a limited impact in rare species where ranges are extremely restricted (e.g. *D. rabepierrei*) as there is no scope for geographical structuring of genetic diversity. Pooled sampling, where there are a small number of representatives in each population, as in *S. decussilvae*, *R. robustior* and *D. procumbens*, improves reliability but can also lead to spurious inferences of population decline,

particularly if migration rates among populations are very low [57]. From a conservation perspective, it is also key to identify species that are becoming fragmented and developing population structure due to anthropogenic activities. Similarly, a rare, highly structured species would still be in need of substantial conservation effort, so ultimately the results will work towards the same goal, even if the relative effect of different phenomena on estimated population size decline remains unclear.

It is important to note that coalescent estimation of population sizes calculated as it is here can only determine demographic patterns for the population that the sampled individuals represent, which may not reflect the species as a whole but instead processes happening at the local scale. As the majority of species were collected in the fragmented forests of Andasibe (figure 1), it is possible that these declines are only pertinent to the Andasibe populations (although high $N_e$ estimates do support a more widespread population decline). This region experienced very rapid deforestation that has recently been limited by community-based conservation efforts [58] and our results highlight the high levels of population decline that have happened even in currently well-managed areas. Further work could examine patterns in other parts of species' ranges, where possible, to identify whether demographic patterns are similar. As habitat fragmentation is an important concept for species conservation, future work should aim to estimate and incorporate levels of population structure and rates of migration among populations into demographic modelling to improve the applicability of approaches such as the one used in this study.

## (g) Genomic data can help target conservation effort

Our ddRADseq approach is a relatively cheap and accessible way to rapidly assess past demographic changes with a handful of specimens. It holds great potential to screen species at a much higher rate than is possible with observation-based red listing assessments. At present, there are too many potentially at-risk species and limited resources in terms of time, money and scientific expertise for the species of interest for a detailed conservation assessment at the level of quality required for the IUCN redlist. One way our approach might be useful is to identify species that are in urgent need of more thorough assessment so that other types of conservation efforts are better placed. Likewise, a species that has recently become rare might benefit from a breeding and reintroduction programme to increase its numbers, while a micro-endemic species that has always been rare would not benefit from this. Our approach has the power to detect this historical difference and could ensure effort is focused on species where narrow ranges and small population sizes are not biologically constrained but the result of human action. This would have the potential to change how conservation management is conducted, with substantial and fast results.

Our approach is widely applicable—a single protocol was successful for species from seven different plant families. Demographic trends can be inferred using just a few individuals with large numbers of markers [37], where previous approaches with small numbers of markers (e.g. microsatellites or chloroplast DNA) fall short. This makes our methodology ideal for studying rare and difficult-to-access species, many of which may be the most threatened and in need of assessment. Effective population sizes estimated using coalescent methods are not proxies for the variance effective size often used in conservation efforts

[59], but changes in $N_e$ are informative regarding recent demographic changes, as is shown here. Further work is required to establish the nature of the relationship between the population declines observed in EBSP analyses and the extinction risk of these species, and to determine what role these types of analyses can play in red listing efforts.

## 4. Conclusion

The effects of the current climate crisis and human influences on biodiversity represent one of the most critical issues facing the planet, yet the scale of biodiversity loss is not fully understood. Current IUCN conservation assessment methods are reliant on observed declines in species population and range sizes over the last 100 years, but for the majority of species, this information is not available. We applied next-generation sequencing and population genetic methods to reveal a historical demographic decline in the rare endemic flora of Madagascar. We show that that human influences on biodiversity are likely to have driven many species towards extinction and suggest that extinction risk in many groups may currently be underestimated by current methods, despite the rigour with which the assessment process is conducted. Our results suggest that rapid decline has affected 70% of the species we assessed and it seems likely that many of the micro-endemic species in Madagascar have not always been so rare. Instead, we propose that the radical environmental changes that have taken place in Madagascar since the arrival of humans have driven rapid population decline in Madagascan plants. Our approach can predict extinction risk from demographic patterns inferred from genomic data and has the potential to act as an important tool for rapidly assessing the threat status of poorly known species in need of further study and conservation efforts, particularly for tropical flora and fauna.

**Ethics.** All samples were collected and exported with prior informed consent of Madagascan authorities under all necessary permits, including CITES permits where appropriate. Permit numbers can be found in electronic supplementary material, table S1.

**Data accessibility.** Scripts and fasta files for all analyses and simulations are available from the Dryad Digital Repository: https://doi.org/10.5061/dryad12jm63xw9 [60]. Raw sequence data can be found in SRA bioproject PRJNA753779. The data are provided in the electronic supplementary material [61].

**Authors' contributions.** A.S.T.P. conceived and designed the study with input from all authors. A.J.H. generated and analysed the data with contributions from A.S.T.P., S.C. coordinated sample collection by R.R. and F.R. W.J.E., M.R. and W.J.B. provided samples. A.J.H. and A.S.T.P. wrote the manuscript with contributions from all authors.

**Competing interests.** We declare we have no competing interests.

**Funding.** This work was funded by Howard Lloyd Davies legacy and NERC.

**Acknowledgements.** We thank Rhian J. Smith, Simon Creer, Andrew Foote, Aaron Comeault and Kathy Willis for support and comments.

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
