## [Peer Review File · Proceedings of the Royal Society B: Biological Sciences]

Review History

RSPB-2020-2373.R0 (Original submission)

Review form: Reviewer 1

Recommendation

Major revision is needed (please make suggestions in comments)

Scientific importance: Is the manuscript an original and important contribution to its field?

Good

General interest: Is the paper of sufficient general interest?

Good

Quality of the paper: Is the overall quality of the paper suitable?

Good

Is the length of the paper justified?

Yes

Should the paper be seen by a specialist statistical reviewer?

Yes

Do you have any concerns about statistical analyses in this paper? If so, please specify them explicitly in your report.

No

It is a condition of publication that authors make their supporting data, code and materials available - either as supplementary material or hosted in an external repository. Please rate, if applicable, the supporting data on the following criteria.

Is it accessible?

N/A

Is it clear?

N/A

Is it adequate?

N/A

Do you have any ethical concerns with this paper?

No

Comments to the Author

This study addresses the demographical history of Malagasy plants with genomic data, and the question on species rarity over time is a very relevant question in Conservation Biology. Plant endemics are supposedly impacted by an increase of human activities during the last two millennia, and the authors tested population decline during that period. They also suggest that their approach could be used to assess the vulnerability of species for which data are deficient. From genomic data, effective population size (N_e) was estimated and compared to census population size (N_c). A higher N_e than N_c may reflect a rapid, recent population decline. The authors also estimated the demographic history by inferring N_e over time. Their analyses were conducted on several plant models, some with reduced population size and critically endangered, while others seem not be impacted by human activities. Recent population decline signatures were detected in most of species, even in species with low concern such as *Psiadia altissima*. Overall, I consider this work is interesting but based on some strong assumptions that need to be carefully considered.

The method has certainly a potential to investigate the recent population demography and vulnerability of micro-endemic species, but I suspect the method could be impacted by many factors that may be discussed, at least briefly. i) First, the method should be pertinent at the local scale, and thus should not be very useful for assessing the status of plants at the island scale. A comparison of demographic pattern between sites may thus be very informative. The problem is mentioned at line 182, but it should be discussed. ii) The habitat of Malagasy plants is generally heterogeneous, and population connectivity could be affected for many reasons, both human-induced and natural (climate-driven). Many groups of plants belong to species complex, and the native flora of Madagascar is also characterized by many species with limited dispersal capacity. In that context, we could expect that recurrent admixture between taxa may happen after habitat perturbation, or conversely, become suddenly impossible. Such events may have strong impact on the genome, and I'm wondering what could be their consequences on inferences made in the present study. In addition, in plants, and especially woody species, generations can overlap, and may also impact demographic inferences. iii) Lastly, I supposed that all plant models investigated in this study were diploids. Autopolyploids (or segmental allopolyploids) are quite common and may not be appropriate for such studies.

Other remarks.

Line 49. Reference "Schatz, 2009" is not numbered and not in the reference list.

Line 92. The N_e vs N_c comparison between plants and humans is maybe not so relevant. Tree

demography may be impacted by many factors not at play in humans, such as generation overlapping.

Lines 103 and 235. A generation time of 10 years is arbitrary considered. I think the average time for replacing a mature breeding individual could strongly vary among species, but also over time. I also feel this value of 10 years is quite short for the tree species considered in the study.

Line 205-209. On the choice of species, could you precise their ploidy and the presence of congeners occurring in the study area?

References.

- [8] Wilme => Wilmé (and complete the reference, the journal is not given)

- [14] and [33], complete the references (vol and article no)

Figures.

Figures are clear, but a picture is missing for *Psychotria* sp. (Fig. 4).

Line 436. "in in" => "in"

Review form: Reviewer 2

Recommendation

Major revision is needed (please make suggestions in comments)

Scientific importance: Is the manuscript an original and important contribution to its field?

Good

General interest: Is the paper of sufficient general interest?

Excellent

Quality of the paper: Is the overall quality of the paper suitable?

Acceptable

Is the length of the paper justified?

Yes

Should the paper be seen by a specialist statistical reviewer?

No

Do you have any concerns about statistical analyses in this paper? If so, please specify them explicitly in your report.

No

It is a condition of publication that authors make their supporting data, code and materials available - either as supplementary material or hosted in an external repository. Please rate, if applicable, the supporting data on the following criteria.

Is it accessible?

No

Is it clear?

No

Is it adequate?

No

Do you have any ethical concerns with this paper?

Yes

Comments to the Author

This paper presents an interesting and clever method to use the effective population size of a species, and its historical change, to obtain information about the extinction risk of species that are otherwise difficult to assess. This is certainly an important issue, as only a minority of species worldwide and across taxa have been assessed, so there are many more at-risk species than we have currently recorded. Overall, the manuscript is easy to follow and flows well. I am pleased to have had the opportunity to review it.

However, there are some shortcomings to the current manuscript. I have some major and minor comments that I hope the authors will find helpful. Please find these attached in a separate document. Most of the minor comments relate to specific line numbers. Some, while linked to specific line numbers, are also of concern more broadly.

Review form: Reviewer 3

Recommendation

Major revision is needed (please make suggestions in comments)

Scientific importance: Is the manuscript an original and important contribution to its field?

Good

General interest: Is the paper of sufficient general interest?

Good

Quality of the paper: Is the overall quality of the paper suitable?

Acceptable

Is the length of the paper justified?

Yes

Should the paper be seen by a specialist statistical reviewer?

No

Do you have any concerns about statistical analyses in this paper? If so, please specify them explicitly in your report.

No

It is a condition of publication that authors make their supporting data, code and materials available - either as supplementary material or hosted in an external repository. Please rate, if applicable, the supporting data on the following criteria.

Is it accessible?

No

Is it clear?

Yes

Is it adequate?

Yes

Do you have any ethical concerns with this paper?

No

Comments to the Author

In the present study, ddRAD data on 43 individuals representing 10 species are analyzed to estimate present and past effective population size. Furthermore, simulations are run to investigate, if with the available dataset, changes in population size can be detected.

Where census is available, the authors generally find much higher estimates of effective population size than census size. They interpret this observation as evidence for recent drastic population declines. This would suggest that many currently rare endemic species in Madagascar were potentially much more abundant in the past.

This is per se an interesting study once more pointing out the importance to not only consider census but also genetic information when assessing a species extinction risk and demonstrating the potential of low-budget genomic projects in doing so.

However, I am not entirely convinced by the analysis. I regard the information currently provided as not sufficient to support the conclusions and I am not sure if the sampling scheme of only very few individuals per species is enough. Furthermore, a more critical discussion of the EBSP analyses (in particular regarding effects of gene flow and population structure) is needed.

My main concerns are the following:

Ne estimates in general, and Bayesian skyline plots in particular, are sensitive to population structure and gene flow. Locally estimating a larger Ne than the observed census size may also be indicative of gene flow with another population or population structure among sampling sites. Also the shape of skyline plots can be highly influenced by these factors (see for instance Heller et al. 2013). At least some discussion of these caveats is important to add.

In the discussion (l. 178-181) the authors admit, that some of these population size estimates may not really be representative of the entire species. This seems a bit in contradiction with the general conclusion about past abundance of current rare endemic species.

I find the results presented in the main figures (2-4) hard to interpret. The units are not comparable with the Ne and Nc estimates given above. I quickly tried to estimate Ne for figure 2a using the y-axis and the mutation rate given in the methods, but this resulted in Ne values close to 2 mio. I am surely doing something wrong... Please give useful units (plain Ne) and generations or years for time. Also, what explains the general increase of Ne over the time frame analysed?

Furthermore, I think clarification is needed regarding the raw data analysis. For instance, what was the coverage at the loci used for the analyses? Coverage can have important impacts on diversity and hence Ne estimates.

It didn't come clear to me, what is meant by the different "read lengths" (147bp, 380bp, 598bp), how they got obtained (see also my comment for l. 223). Where does the difference among species come from (different sequencing efforts?)?. I also think there could be a correlation of this measure with diversity measure (He, etc), please check.

The raw data analysis was performed per species and therefore each time with only very few samples (2 to 5 samples). I expect that disentangling sequencing errors from real SNPs is very difficult with so few samples while sequencing errors are problematic for Bayesian skyline plots.

The possibility of simply underestimating census size, because some may be difficult to find, should be discussed a bit more.

Specific comments:

I think it would be helpful to have some of the information given in Table S2 in the main text

(Species name, sample number and diversity measurements).

l. 23: some sort of break needed between extinctions and yet?

l. 26: as N_e estimates can be tricky, I suggest to mention in the abstract, that this study used ddRAD.

l. 28-29: or over/underestimation of one of the two...

l. 32: does this refer to simulated data? and declines of what?

l. 81 decreased in area?

l. 85: what is meant by training data set?

l. 103: as the generation time is important here, some information is needed how it was chosen and how accurate it may be to use the same for all these plants.

l. 105: Could this also be explained by gene flow among populations/species?

Also, a naive question from a non-botanist: how accurate is census size? From the pictures on Figure 2 it seems well possible, that these plants are not very easy to find?

To also give census over time would be informative.

l. 118: Please explain, what is meant by the "sum(indicstors.alltrees)".

l. 125: which endangered species do the authors mean? endangered palm trees or other plants? And are these declines in N_c or N_e ? Is there a figure or table showing this?

l. 164-166: but even in the simulations with 10% increase, there was a small decline?

l. 200-201: this is contradicted by l. 178-181, where the authors admit that some of these population estimates may not represent the entire species.

l. 205: seven plant families?

l. 223: I assume, the 380bp is without the adapters, but 147bp would be much smaller than expected from the library size (~500bp)? Also, I see how overlapping reads can be concatenated (or merged), but to merge the 75bp, the introduction of Ns would have been necessary with this library size?

l. 224: Aligned to the concatenated tags? I read between lines that this was a denovo analysis, ie. no reference genomes used? If yes, please specify for people who, as me, never used Stacks. Could you give somewhere a range of approximate genome sizes?

l. 224: How well does Stacks perform with such low sample sizes per species? I would assume that SNP calling (disentangling SNPs from sequencing errors) is tricky with few samples. Sequencing errors are, to my knowledge problematic for Bayesian skyline plots.

l. 230: ...output of from the...

l. 230: how were diploids (or even polyploids?) handled? Was there one or more fasta sequences per individual?

l. 239: I see that the authors cite another paper for this so my question is probably naive. Still, by focusing on regions with high diversity, isn't there the risk of overestimating global diversity and hence N_e ?

l. 241: why was only a very small subset of the loci chosen?

l. 244: what is meant by SNP class?

l. 247: were the replicate runs performed with different sets of 50 loci (l. 241)?

l. 250: does this mean that for most species, only 50 loci were available?

l. 255: is the generation time for all these different species really always about 10 or were the authors thinking about a particular species?

l. 257: ...estimates for? *D. procumbens*...

l. 259: what is meant by SNP category here?

Figure 2, y axis and IUCN status are difficult to read.

What do the bp numbers mean?

l. 417/418: where can this be found?

Could the time be shown in generations or years?

Figure 3: what do the different lines per panel represent? The legend in d) represents the four different panels I assume? It would be very interesting to see, what scenario led to an increase and then decrease of pop size. It may be more intuitive to compare this figure with the empirical data, if also here a logarithmic scale was shown. Or is then no change visible anymore?

Decision letter (RSPB-2020-2373.R0)

30-Oct-2020

Dear Dr Helmstetter,

We have now received referees' reports on your manuscript RSPB-2020-2373 entitled "The demographic history of Madagascan micro-endemics: have rare species always been rare?".

The referees, Associate Editor and I are all agreed that this a valuable study that could potentially be a very interesting paper, but several substantial concerns have been raised. The paper has therefore been rejected for publication in Proceedings B in its current form. However we would be happy to consider a resubmission, provided the comments of the referees and the Associate Editor are fully addressed.

The referees and AE have provided detailed comments, which are listed below. In addition to the detailed issues to be addressed, please pay attention to the three following components: you need to make clear the relevance of the paper for a general biology journal, rather than a more specialised conservation publication; please provide information on the relevant Malagasy permits; and please ensure that the archived data are publicly accessible.

Please note that this is not a provisional acceptance. The resubmission will be treated as a new manuscript. However, we will approach the same reviewers if they are available and it is deemed appropriate to do so by the Editor. Please note that resubmissions must be submitted within six months of the date of this email. In exceptional circumstances, extensions may be possible if agreed with the Editorial Office. Manuscripts submitted after this date will be automatically rejected.

If you do choose to resubmit your manuscript, please upload the following:

Thank you for submitting to Proceedings B, and I hope all is well with you and all your co-authors in this difficult year.

Yours sincerely,
 Professor Loeske Kruuk
 Editor
 mailto: proceedingsb@royalsociety.org

Associate Editor
 Board Member: 1
 Comments to Author:

Many thanks for submitting your work on the use of genomic data to infer the effective population size of 10 Madagascan endemics. Your manuscript has now been reviewed by three experts, and all agree that your finding that effective population sizes are higher than current census sizes is fascinating. I fully agree with this assessment. However, at the same time they all raise concerns regarding the methods, the interpretation and implications of the results, and the generality of these findings. I refer you to their thorough and constructive reviews for more details, but will briefly summarise some of the issues raised below.

Although reviewer 1 agrees that the method outlined has potential, they are concerned that some of the assumptions may not be justified, and that there may be other processes that could generate similar patterns. Reviewer 2 expands on this and, among others, highlights the need for more discussion of the effect of mutation rate and generation time on the inferred timescale of the population declines. Similarly, reviewer 3 expresses some concern regarding the robustness of the results given the relatively small sample sizes for each species and the known sensitivity of these analyses to gene flow and population structure.

In addition to these comments, I have some concerns about the relatively small number of species involved. Indeed, although the fact that there is no evidence for a recent decline in the two near-threatened palm species, but there is in the two endangered species, is in line with predictions, the sample size for this analysis is four. Convincingly showing that EBS analysis can predict conservation status would require a much larger number of species.

To summarise, this manuscript presents an interesting method that has the potential to change how we assess the conservation status of species/populations. Furthermore, it provides some interesting insights into the demographic history of a number of Madagascan endemics. However, in its current form it is difficult to assess the robustness of the main findings. Furthermore, I am concerned that the manuscript may not be of sufficient interest to the broader readership of Proceedings B: One half of the manuscript is very methodological and hence it be more appropriate for a journal like Molecular Ecology Resources or maybe Methods in Ecology and Evolution. The other half is relevant to the conservation of a specific set of Madagascan species, and thereby more appropriate for a conservation-oriented journal. Having said this, if the method as presented here can be shown to be robust, and there are good reasons to believe that the results obtained for this subset of species are representative of a more general phenomenon, than Proceedings B could be a good fit for this study.

Reviewer(s)' Comments to Author:

Referee: 1

Comments to the Author(s)

This study addresses the demographical history of Malagasy plants with genomic data, and the question on species rarity over time is a very relevant question in Conservation Biology. Plant endemics are supposedly impacted by an increase of human activities during the last two millennia, and the authors tested population decline during that period. They also suggest that their approach could be used to assess the vulnerability of species for which data are deficient. From genomic data, effective population size (N_e) was estimated and compared to census population size (N_c). A higher N_e than N_c may reflect a rapid, recent population decline. The authors also estimated the demographic history by inferring N_e over time. Their analyses were conducted on several plant models, some with reduced population size and critically endangered, while others seem not be impacted by human activities. Recent population decline signatures were detected in most of species, even in species with low concern such as *Psiadia altissima*. Overall, I consider this work is interesting but based on some strong assumptions that need to be carefully considered.

The method has certainly a potential to investigate the recent population demography and vulnerability of micro-endemic species, but I suspect the method could be impacted by many factors that may be discussed, at least briefly. i) First, the method should be pertinent at the local scale, and thus should not be very useful for assessing the status of plants at the island scale. A comparison of demographic pattern between sites may thus be very informative. The problem is mentioned at line 182, but it should be discussed. ii) The habitat of Malagasy plants is generally heterogeneous, and population connectivity could be affected for many reasons, both human-induced and natural (climate-driven). Many groups of plants belong to species complex, and the native flora of Madagascar is also characterized by many species with limited dispersal capacity. In that context, we could expect that recurrent admixture between taxa may happen after habitat perturbation, or conversely, become suddenly impossible. Such events may have strong impact on the genome, and I'm wondering what could be their consequences on inferences made in the present study. In addition, in plants, and especially woody species, generations can overlap, and may also impact demographic inferences. iii) Lastly, I supposed that all plant models investigated in this study were diploids. Autopolyploids (or segmental allopolyploids) are quite common and may not be appropriate for such studies.

Other remarks.

Line 49. Reference "Schatz, 2009" is not numbered and not in the reference list.

Line 92. The N_e vs N_c comparison between plants and humans is maybe not so relevant. Tree demography may be impacted by many factors not at play in humans, such as generation overlapping.

Lines 103 and 235. A generation time of 10 years is arbitrary considered. I think the average time for replacing a mature breeding individual could strongly vary among species, but also over time. I also feel this value of 10 years is quite short for the tree species considered in the study.

Line 205-209. On the choice of species, could you precise their ploidy and the presence of congeners occurring in the study area?

References.

- [8] Wilme => Wilmé (and complete the reference, the journal is not given)
- [14] and [33], complete the references (vol and article no)

Figures.

Figures are clear, but a picture is missing for *Psychotria* sp. (Fig. 4).

Line 436. "in in" => "in"

Referee: 2

Comments to the Author(s)

This paper presents an interesting and clever method to use the effective population size of a species, and its historical change, to obtain information about the extinction risk of species that are otherwise difficult to assess. This is certainly an important issue, as only a minority of species worldwide and across taxa have been assessed, so there are many more at-risk species than we have currently recorded. Overall, the manuscript is easy to follow and flows well. I am pleased to have had the opportunity to review it.

However, there are some shortcomings to the current manuscript. I have some major and minor comments that I hope the authors will find helpful. Please find these attached in a separate document. Most of the minor comments relate to specific line numbers. Some, while linked to specific line numbers, are also of concern more broadly.

Referee: 3

Comments to the Author(s)

In the present study, ddRAD data on 43 individuals representing 10 species are analyzed to estimate present and past effective population size. Furthermore, simulations are run to investigate, if with the available dataset, changes in population size can be detected.

Where census is available, the authors generally find much higher estimates of effective population size than census size. They interpret this observation as evidence for recent drastic population declines. This would suggest that many currently rare endemic species in Madagascar were potentially much more abundant in the past.

This is per se an interesting study once more pointing out the importance to not only consider census but also genetic information when assessing a species extinction risk and demonstrating the potential of low-budget genomic projects in doing so.

However, I am not entirely convinced by the analysis. I regard the information currently provided as not sufficient to support the conclusions and I am not sure if the sampling scheme of only very few individuals per species is enough. Furthermore, a more critical discussion of the EBSP analyses (in particular regarding effects of gene flow and population structure) is needed.

My main concerns are the following:

Ne estimates in general, and Bayesian skyline plots in particular, are sensitive to population structure and gene flow. Locally estimating a larger Ne than the observed census size may also be indicative of gene flow with another population or population structure among sampling sites.

Also the shape of skyline plots can be highly influenced by these factors (see for instance Heller et al. 2013). At least some discussion of these caveats is important to add.

In the discussion (l. 178-181) the authors admit, that some of these population size estimates may not really be representative of the entire species. This seems a bit in contradiction with the general conclusion about past abundance of current rare endemic species.

I find the results presented in the main figures (2-4) hard to interpret. The units are not comparable with the N_e and N_c estimates given above. I quickly tried to estimate N_e for figure 2a using the y-axis and the mutation rate given in the methods, but this resulted in N_e values close to 2 mio. I am surely doing something wrong... Please give useful units (plain N_e) and generations or years for time. Also, what explains the general increase of N_e over the time frame analysed?

Furthermore, I think clarification is needed regarding the raw data analysis. For instance, what was the coverage at the loci used for the analyses? Coverage can have important impacts on diversity and hence N_e estimates.

It didn't come clear to me, what is meant by the different "read lengths" (147bp, 380bp, 598bp), how they got obtained (see also my comment for l. 223). Where does the difference among species come from (different sequencing efforts?)?. I also think there could be a correlation of this measure with diversity measure (H_e , etc), please check.

The raw data analysis was performed per species and therefore each time with only very few samples (2 to 5 samples). I expect that disentangling sequencing errors from real SNPs is very difficult with so few samples while sequencing errors are problematic for Bayesian skyline plots.

The possibility of simply underestimating census size, because some may be difficult to find, should be discussed a bit more.

Specific comments:

I think it would be helpful to have some of the information given in Table S2 in the main text (Species name, sample number and diversity measurements).

l. 23: some sort of break needed between extinctions and yet?

l. 26: as N_e estimates can be tricky, I suggest to mention in the abstract, that this study used ddRAD.

l. 28-29: or over/underestimation of one of the two...

l. 32: does this refer to simulated data? and declines of what?

l. 81 decreased in area?

l. 85: what is meant by training data set?

l. 103: as the generation time is important here, some information is needed how it was chosen and how accurate it may be to use the same for all these plants.

l. 105: Could this also be explained by gene flow among populations/species?

Also, a naive question from a non-botanist: how accurate is census size? From the pictures on Figure 2 it seems well possible, that these plants are not very easy to find?
To also give census over time would be informative.

l. 118: Please explain, what is meant by the "sum(indicstors.alltrees)".

l. 125: which endangered species do the authors mean? endangered palm trees or other plants? And are these declines in N_c or N_e ? Is there a figure or table showing this?

l. 164-166: but even in the simulations with 10% increase, there was a small decline?

- l. 200-201: this is contradicted by l. 178-181, where the authors admit that some of these population estimates may not represent the entire species.
- l. 205: seven plant families?
- l. 223: I assume, the 380bp is without the adapters, but 147bp would be much smaller than expected from the library size (~500bp)? Also, I see how overlapping reads can be concatenated (or merged), but to merge the 75bp, the introduction of Ns would have been necessary with this library size?
- l. 224: Aligned to the concatenated tags? I read between lines that this was a denovo analysis, ie. no reference genomes used? If yes, please specify for people who, as me, never used Stacks. Could you give somewhere a range of approximate genome sizes?
- l. 224: How well does Stacks perform with such low sample sizes per species? I would assume that SNP calling (disentangling SNPs from sequencing errors) is tricky with few samples. Sequencing errors are, to my knowledge problematic for Bayesian skyline plots.
- l. 230: ...output of from the...
- l. 230: how were diploids (or even polyploids?) handled? Was there one or more fasta sequences per individual?
- l. 239: I see that the authors cite another paper for this so my question is probably naive. Still, by focusing on regions with high diversity, isn't there the risk of overestimating global diversity and hence N_e ?
- l. 241: why was only a very small subset of the loci chosen?
- l. 244: what is meant by SNP class?
- l. 247: were the replicate runs performed with different sets of 50 loci (l. 241)?
- l. 250: does this mean that for most species, only 50 loci were available?
- l. 255: is the generation time for all these different species really always about 10 or were the authors thinking about a particular species?
- l. 257: ...estimates for? *D. procumbens*...
- l. 259: what is meant by SNP category here?

Figure 2, y axis and IUCN status are difficult to read.

What do the bp numbers mean?

l. 417/418: where can this be found?

Could the time be shown in generations or years?

Figure 3: what do the different lines per panel represent? The legend in d) represents the four different panels I assume? It would be very interesting to see, what scenario led to an increase and then decrease of pop size. It may be more intuitive to compare this figure with the empirical data, if also here a logarithmic scale was shown. Or is then no change visible anymore?

Author's Response to Decision Letter for (RSPB-2020-2373.R0)

See Appendix A.

RSPB-2021-0957.R0

Review form: Reviewer 2

Recommendation

Reject – article is not of sufficient interest (we will consider a transfer to another journal)

Scientific importance: Is the manuscript an original and important contribution to its field?

Good

General interest: Is the paper of sufficient general interest?

Acceptable

Quality of the paper: Is the overall quality of the paper suitable?

Acceptable

Is the length of the paper justified?

Yes

Should the paper be seen by a specialist statistical reviewer?

No

Do you have any concerns about statistical analyses in this paper? If so, please specify them explicitly in your report.

Yes

It is a condition of publication that authors make their supporting data, code and materials available - either as supplementary material or hosted in an external repository. Please rate, if applicable, the supporting data on the following criteria.

Is it accessible?

No

Is it clear?

N/A

Is it adequate?

No

Do you have any ethical concerns with this paper?

Yes

Comments to the Author

I appreciate the authors' efforts to respond to all three reviewers, especially given the detailed comments. In addition, the authors provide additional simulations to show that mis-specification of the generation time does not change the comparisons between N_e and N_c , and provide discussion about both the effects of mutation rate and population structure. They show that their

sample size is adequate for the EBSP analyses they perform, and provide confidence intervals for the N_e they calculate.

However, I have some remaining concerns, and some new concerns brought up by the additional information provided in response to my and the other reviewer's comments.

First, the authors clarify in their response that this work is not intended to "change/rewrite" IUCN assessment criteria, but "just add more information" for current and novel assessments. While more information is always useful, some concrete examples of how this additional information might be used would be beneficial here to show the general use of this method. As the authors point out in their response, there are indeed multiple criteria and subcriteria for assessment, simply adding another piece of information does not seem of broad enough interest. One possibility might be for direct conservation management. A species that has recently become rare might benefit from a breeding and reintroduction program to increase its numbers, while a microendemic species that has always been rare would not benefit from this, and efforts could more usefully be put toward protecting its existing habitat and individuals. This would have the potential to change how conservation management is conducted, with substantial and fast results.

I very much appreciate the additional simulations the authors performed, since these can be quite time consuming. Some of these simulations show that a population expansion of 10% can give an EBSP that looks similar to decline of about the same magnitude, and the estimates of the magnitude of decline are vary variable (eg. -0.21 to -0.80 for a 10% decline with 380 bp reads, -0.53 to -0.96 for a 10% decline with 147 bp reads) so I now wonder how reliable the EBSP conclusions of population declines and their magnitudes are. For example, the decline estimated in *Satranala decussilvae*, concluded to have undergone a 78% population decline, falls within the range of declines simulated for simulations of 10% expansion, 10% decline, and 50% decline.

Regarding the additional information provided in Supplemental Table 3, the 95% confidence limits of the estimates of θ include 0 in 7 of the 10 species, but the listed N_e estimates do not. The formula for calculating N_e from θ is $N_e = \theta / 4\mu$, so when $\theta = 0$, $N_e = 0$ as well, unless there are typos for these seven estimates of the lower CI or some additional calculations are going on other than those described in the ThetaMater documentation. If the true CIs do include 0 (or indeed the N_c of that species), then reported differences between N_e and N_c are not significant.

Lastly, while the authors do comment on how population structure could affect these patterns, I find their discussion to be insufficient to the potential for confounding. They cite Heller et al. 2013, who found that in structured populations, EBSPs with only local samples of a population, as well as pooled samples, can exhibit false signals of population decline (Fig. 1 A-E), though the effect in pooled samples is more pronounced with very low migration, and scattered samples do not. They define local samples as all samples from a single deme, pooled samples as multiple samples from a subset of demes, and scattered samples as one sample from each deme in the population. The authors comment that their locally-sampled species may therefore be subject to these false signals, but not the palms which were sampled from multiple demes. Unless I am misunderstanding the authors' sampling scheme in this work, the sampling scheme of the palms may more closely match Heller et al's definition of "pooled" samples, especially for *Satranala decussilvae* (2 samples each from 2 populations).

Furthermore, Mazet et al. 2016 (Heredity) show that "any demographic model with structure will necessarily be interpreted as a series of changes in population size by inference methods ignoring structure." While I agree with the authors that investigating gene flow among populations and determining if and how it would affect estimates of demographic change and N_e would be an interesting avenue for a comprehensive future study, I maintain that some analytical treatment is necessary to show the general utility of this method. This is especially important since populations of conservation concern are frequently restricted to habitat fragments, and therefore

may be subject to strong, anthropogenic population structure, in a time frame comparable to the time frame of the declines being investigated.

Review form: Reviewer 3

Recommendation

Major revision is needed (please make suggestions in comments)

Scientific importance: Is the manuscript an original and important contribution to its field?

Good

General interest: Is the paper of sufficient general interest?

Good

Quality of the paper: Is the overall quality of the paper suitable?

Acceptable

Is the length of the paper justified?

Yes

Should the paper be seen by a specialist statistical reviewer?

No

Do you have any concerns about statistical analyses in this paper? If so, please specify them explicitly in your report.

No

It is a condition of publication that authors make their supporting data, code and materials available - either as supplementary material or hosted in an external repository. Please rate, if applicable, the supporting data on the following criteria.

Is it accessible?

N/A

Is it clear?

Yes

Is it adequate?

Yes

Do you have any ethical concerns with this paper?

No

Comments to the Author

The authors have responded to most of my comments and the manuscript has improved. However, I remain with some rather major comments.

I now see why my trial to estimate N_e from the EBSP results didn't work. I was using the wrong mutation rate as these estimates are based on highly variable RAD tags. When comparing absolute values of N_e and N_c , the mutation rate is a crucial player. The authors just explain, why they didn't have a better estimate, but they definitely need to show, how robust their main results are to different mutation rates and properly discuss this issue.

I was surprised to read in the response to me that gene flow and population structure are not relevant, if at the same time, the authors did add it to the discussion as response to reviewer 1. I am glad they added some discussion on this. What I am missing a bit, is that a few clustered populations (hence population structure) could lead to an overestimation of N_e , also in the case of the direct estimation using thetamateR.

Also, declines in coalescence-based N_e estimates in the recent past is often observed and can simply be due to local effects. Ancestors from the close past are more likely to be from nearby and hence more likely to be genetically similar.

The authors have improved the presentation of the raw data analysis. An information, which would help readers, who don't know stacks, is if it also retains monomorphic loci. I believe this is important because of the mutation rate used for the thetamateR N_e estimates. I assume the mutation rate used is a genome-wide estimate, so including invariants sites would be correct?

Concerning the raw data analysis, I am still missing what kind of filtering was performed. Were singletons kept? Was there any filtering for heterozygosity? De novo analysis of RAD data has the risk to lead to paralogous sites. This could inflate N_e estimates.

Related: I still find the use of only highly variable RAD tags to infer past N_e a little scary. 3 or more SNPs per locus seems like a lot and the risk to include paralogous sequences high. Please give the heterozygosity distribution at RAD tags used for the N_e estimates and for the EBSP analyses.

In the simulations, the expansion is not detected. The authors explain this to me by lack of coalescence events in the past. This means that their method can only show declines, even in a scenario of expansion. This should at least be mentioned somewhere in the text.

Related to this: is the number of replicates displayed in Figure 3a the same as in b-d?

And rather a comment to the author response regarding sample size: Of course low sample size is a common issue in conservation. To prove a principle, it could then be better to do such analysis using a more abundant species and infer population size by down-/resampling.

l. 152-154: Wouldn't it be correct to reject constant size if there was a change in population size, even if it wasn't severe?

l. 314: with pipeline above, the entire pipeline from SNP calling to N_e estimates is meant?

Table S2: Please indicate number of variable sites as well

Decision letter (RSPB-2021-0957.R0)

07-Jun-2021

Dear Dr Helmstetter:

Thank you for submitting a revised version of this manuscript, which has now been peer reviewed and the reviews have been assessed by an Associate Editor. The reviewers' comments (not including confidential comments to the Editor) and the comments from the Associate Editor are included at the end of this email for your reference, and as you will see, all three have provided very thorough reviews. Whilst they all appreciate the work that you have put in to the revision, there are nevertheless some major concerns with this version of the manuscript. I would therefore like to invite you to revise your manuscript to address these issues.

Research ethics:

Use of animals and field studies:

It is a condition of publication that you make available the data and research materials supporting the results in the article (<https://royalsociety.org/journals/authors/author-guidelines/#data>). Datasets should be deposited in an appropriate publicly available repository and details of the associated accession number, link or DOI to the datasets must be included in the Data Accessibility section of the article (<https://royalsociety.org/journals/ethics-policies/data-sharing-mining/>). Reference(s) to datasets should also be included in the reference list of the article with DOIs (where available).

Please submit a copy of your revised paper within three weeks. If we do not hear from you within this time your manuscript will be rejected. If you are unable to meet this deadline please let us know as soon as possible, as we may be able to grant a short extension.

Best wishes,
Professor Loeske Kruuk
mailto:proceedingsb@royalsociety.org

Associate Editor
Comments to Author:

Many thanks for submitting a revised version of your manuscript. As you will see below, your revision has been reviewed by two reviewers, both of which also reviewed the original submission.

Like both reviewers, I really appreciate the effort that has gone into this revision. In particular the power analyses are an immensely valuable addition, and we all agree that these go a long way in addressing concerns regarding the robustness of your results. Nevertheless, you will find that both reviewers remain relatively critical of some of your results and their interpretation.

For example, Reviewer 1 argues that the recent declines in population size in may be overestimates, and that population structure may have inflated estimates of N_e . Although this is briefly discussed in the revision, it would be worth expanding this section and/or making it more prominent (also see my comment regarding the structure below). Reviewer 1 also still misses some important information on how the raw data was analysed, and the filtering in particular.

Reviewer 2 also remains rather critical and raises a number of further concerns regarding the interpretation of the simulation results and the high uncertainty around some of the estimates. Furthermore, like Reviewer 1, they remain concerned about the potential effect of population structure.

Although several of the issues raised are pretty substantial, I also believe that many of these could be addressed in a further revision that either removes the concerns raised, or directly acknowledges them. Almost all studies will have some caveats and/or rely on potentially oversimplistic assumptions, but as long as they are clearly acknowledged and it is made clear what can learn from this study despite its limitations, they are a valuable contribution to the literature.

This brings me to my concern that your manuscript may be too methodological/specialist for a journal like *Proceeding B*. Indeed, the addition of the simulations and the expanded discussion of the robustness of the results has resulted in a manuscript that is arguably more technical and method-oriented than the previous version. Having said this, I also appreciate the open and transparent discussion of what we can and cannot learn from analyses such as these. Furthermore, I found that the introduction makes a convincing argument for why we need alternative methods to assess population trends and to infer extinction risk, and for why genomic methods may provide such an alternative. In fact, your significance and general interest statement in the cover are very strong and well-written, and I think it would be a pity not to incorporate them into the manuscript itself.

This brings me to the current structure of your manuscript. Your decision to combine Results and Discussion and to present these before the Methods is to some degree a matter of taste, but given its large methodological component, would it not be more appropriate to make the Methods section more prominent by moving it forward, instead of 'hiding' it at the end of the manuscript?

Indeed, I think there still is some room for improvement when it comes to the structure of the manuscript in general: The Introduction currently consists of only two, very long and rather dense paragraphs. Would you be able to split these into multiple shorter paragraphs? The same is true for several paragraphs in the Results and Discussion section. I think this should be pretty straightforward and it would significantly improve readability. I also think that several sections in the Results and Discussion would be more at home in the Introduction, such as the general explanation of what we can learn from the comparison of effective and census population size and previous examples of this approach.

Currently the Introduction is one manuscript page, and the Results and Discussion covers five pages that include everything from methodological details to general conclusions. I suspect that adopting a more 'traditional' structure, shorter paragraphs and maybe even some subheadings, will make your manuscript accessible to a wider readership: It will allow people to read only those sections relevant to their interests, and at the same time important details are less likely to get snowed under.

Finally, although the mention of permits etc. for collection and export is an important addition, it currently is extremely generic. I think this should really be accompanied by some form of proof, e.g. by listing permit numbers and providing details on who provided the permits. You give more details in the response to the reviewers, but this (and ideally more) should be included in the manuscript itself. Also, Figure 1 could be improved: The information content is very low and I wonder if it is needed as part of the main text at all. If it remains, is there an alternative to colour-coding the species? Different symbols or letters? Arrows?

Reviewer(s)' Comments to Author:

Referee: 3

Comments to the Author(s).

The authors have responded to most of my comments and the manuscript has improved. However, I remain with some rather major comments.

I now see why my trial to estimate N_e from the EBSP results didn't work. I was using the wrong mutation rate as these estimates are based on highly variable RAD tags. When comparing absolute values of N_e and N_c , the mutation rate is a crucial player. The authors just explain, why they didn't have a better estimate, but they definitely need to show, how robust their main results are to different mutation rates and properly discuss this issue.

I was surprised to read in the response to me that gene flow and population structure are not relevant, if at the same time, the authors did add it to the discussion as response to reviewer 1. I am glad they added some discussion on this. What I am missing a bit, is that a

few clustered populations (hence population structure) could lead to an overestimation of N_e , also in the case of the direct estimation using *thetamater*.

Also, declines in coalescence-based N_e estimates in the recent past is often observed and can simply be due to local effects. Ancestors from the close past are more likely to be from nearby and hence more likely to be genetically similar.

The authors have improved the presentation of the raw data analysis. An information, which would help readers, who don't know stacks, is if it also retains monomorphic loci. I believe this is important because of the mutation rate used for the *thetamater* N_e estimates. I assume the mutation rate used is a genome-wide estimate, so including invariants sites would be correct?

Concerning the raw data analysis, I am still missing what kind of filtering was performed. Were singletons kept? Was there any filtering for heterozygosity? De novo analysis of RAD data has the risk to lead to paralogous sites. This could inflate N_e estimates.

Related: I still find the use of only highly variable RAD tags to infer past N_e a little scary. 3 or more SNPs per locus seems like a lot and the risk to include paralogous sequences high. Please give the heterozygosity distribution at RAD tags used for the N_e estimates and for the EBSF analyses.

In the simulations, the expansion is not detected. The authors explain this to me by lack of coalescence events in the past. This means that their method can only show declines, even in a scenario of expansion. This should at least be mentioned somewhere in the text.

Related to this: is the number of replicates displayed in Figure 3a the same as in b-d?

And rather a comment to the author response regarding sample size: Of course low sample size is a common issue in conservation. To prove a principle, it could then be better to do such analysis using a more abundant species and infer population size by down-/resampling.

l. 152-154: Wouldn't it be correct to reject constant size if there was a change in population size, even if it wasn't severe?

l. 314: with pipeline above, the entire pipeline from SNP calling to N_e estimates is meant?

Table S2: Please indicate number of variable sites as well

Referee: 2

Comments to the Author(s).

I appreciate the authors' efforts to respond to all three reviewers, especially given the detailed comments. In addition, the authors provide additional simulations to show that mis-specification of the generation time does not change the comparisons between N_e and N_c , and provide discussion about both the effects of mutation rate and population structure. They show that their sample size is adequate for the EBSF analyses they perform, and provide confidence intervals for the N_e s they calculate.

However, I have some remaining concerns, and some new concerns brought up by the additional information provided in response to my and the other reviewer's comments.

First, the authors clarify in their response that this work is not intended to "change/rewrite" IUCN assessment criteria, but "just add more information" for current and novel assessments. While more information is always useful, some concrete examples of how this additional information might be used would be beneficial here to show the general use of this method. As the authors point out in their response, there are indeed multiple criteria and subcriteria for assessment, simply adding another piece of information does not seem of broad enough interest. One possibility might be for direct conservation management. A species that has recently become rare might benefit from a breeding and reintroduction program to increase its numbers, while a

microendemic species that has always been rare would not benefit from this, and efforts could more usefully be put toward protecting its existing habitat and individuals. This would have the potential to change how conservation management is conducted, with substantial and fast results.

I very much appreciate the additional simulations the authors performed, since these can be quite time consuming. Some of these simulations show that a population expansion of 10% can give an EBSP that looks similar to decline of about the same magnitude, and the estimates of the magnitude of decline are vary variable (eg. -0.21 to -0.80 for a 10% decline with 380 bp reads, -0.53 to -0.96 for a 10% decline with 147 bp reads) so I now wonder how reliable the EBSP conclusions of population declines and there magnitudes are. For example, the decline estimated in *Satranala decussilvae*, concluded to have undergone a 78% population decline, falls within the range of declines simulated for simulations of 10% expansion, 10% decline, and 50% decline.

Regarding the additional information provided in Supplemental Table 3, the 95% confidence limits of the estimates of theta include 0 in 7 of the 10 species, but the listed N_e estimates do not. The formula for calculating N_e from θ is $N_e = \theta / 4\mu$, so when $\theta = 0$, $N_e = 0$ as well, unless there are typos for these seven estimates of the lower CI or some additional calculations are going on other than those described in the ThetaMater documentation. If the true CIs do include 0 (or indeed the N_c of that species), then reported differences between N_e and N_c are not significant.

Lastly, while the authors do comment on how population structure could affect these patterns, I find their discussion to be insufficient to the potential for confounding. They cite Heller et al. 2013, who found that in structured populations, EBSPs with only local samples of a population, as well as pooled samples, can exhibit false signals of population decline (Fig. 1 A-E), though the effect in pooled samples is more pronounced with very low migration, and scattered samples do not. They define local samples as all samples from a single deme, pooled samples as multiple samples from a subset of demes, and scattered samples as one sample from each deme in the population. The authors comment that their locally-sampled species may therefore be subject to these false signals, but not the palms which were sampled from multiple demes. Unless I am misunderstanding the authors' sampling scheme in this work, the sampling scheme of the palms may more closely match Heller et al's definition of "pooled" samples, especially for *Satranala decussilvae* (2 samples each from 2 populations).

Furthermore, Mazet et al. 2016 (Heredity) show that "any demographic model with structure will necessarily be interpreted as a series of changes in population size by inference methods ignoring structure." While I agree with the authors that investigating gene flow among populations and determining if and how it would affect estimates of demographic change and N_e would be an interesting avenue for a comprehensive future study, I maintain that some analytical treatment is necessary to show the general utility of this method. This is especially important since populations of conservation concern are frequently restricted to habitat fragments, and therefore may be subject to strong, anthropogenic population structure, in a time frame comparable to the time frame of the declines being investigated.

Author's Response to Decision Letter for (RSPB-2021-0957.R0)

See Appendix B.

Decision letter (RSPB-2021-0957.R1)

25-Aug-2021

Dear Dr Helmstetter

I am pleased to inform you that your manuscript entitled "The demographic history of Madagascan micro-endemics: have rare species always been rare?" has been accepted for publication in Proceedings B.

Data Accessibility section

Open Access

You are invited to opt for Open Access, making your freely available to all as soon as it is ready for publication under a CCBY licence. Our article processing charge for Open Access is £1700. Corresponding authors from member institutions (<http://royalsocietypublishing.org/site/librarians/allmembers.xhtml>) receive a 25% discount to these charges. For more information please visit <http://royalsocietypublishing.org/open-access>.

Paper charges

Sincerely,

Professor Loeske Kruuk

Associate Editor:

Board Member

Comments to Author:

Dear Dr Helmstetter and co-authors,

Many thanks for resubmitting your manuscript to Proceedings B, and for taking on board a second series of comments and suggestions. I really appreciate the effort you have invested in in this revision, which I think addresses all issues raised in a satisfactory manner.

All things considered, I think this manuscript makes a convincing case for the incorporation of genomic data in conservation assessments, while carefully discussing potential limitations. At the same time it provides a fascinating insight into the population dynamics of a number of endemic Madagascan plant and its drivers. Therefore I believe that your manuscript will make a significant contribution to the literature and that it will appeal to a broad readership.

Thanks again and with kind regards,

Erik Postma

Appendix A

Associate Editor

Board Member: 1

Comments to Author:

Many thanks for submitting your work on the use of genomic data to infer the effective population size of 10 Madagascan endemics. Your manuscript has now been reviewed by three experts, and all agree that your finding that effective population sizes are higher than current census sizes is fascinating. I fully agree with this assessment. However, at the same time they all raise concerns regarding the methods, the interpretation and implications of the results, and the generality of these findings. I refer you to their thorough and constructive reviews for more details, but will briefly summarise some of the issues raised below.

Although reviewer 1 agrees that the method outlined has potential, they are concerned that some of the assumptions may not be justified, and that there may be other processes that could generate similar patterns.

The assumptions we have made are relevant to the questions we pose, particularly in relation to the study of rare species, which we develop upon in new passages in our paper (L160). We now consider the potential of other processes, such as gene flow or generation time, to affect our inferences and why their effect is likely limited in our study (L164).

Reviewer 2 expands on this and, among others, highlights the need for more discussion of the effect of mutation rate and generation time on the inferred timescale of the population declines.

We have now added a discussion of the potential effects of mutation rate and generation time. We have added new simulations for events both younger (100 generations) and older (500 generations) than the original 250 generations (2500 years) that account for these effects on our ability to infer population size change and show that our results are robust to this uncertainty.

Similarly, reviewer 3 expresses some concern regarding the robustness of the results given the relatively small sample sizes for each species and the known sensitivity of these analyses to gene flow and population structure.

We understand the concern of the reviewer but there are several reasons why it is not so relevant to our study:

- Population structure and gene flow would give signals of expansion due to the increased number of haplotypes. In which case any meaningful effect on our conclusions would only apply to the stable/expanding species so that, if anything, species are doing less well than our models indicate.
- Many rare species are found only in a single or a few geographically clustered populations (e.g. *S. decussilvae*, *D. rabepierrei*), which limits the potential for gene flow and structure.

- Low sample sizes cannot be avoided when working with rare species, which is the essence of our study, and what makes it useful and of wider interest. Understanding the limitations and effectiveness of inference with low sample sizes is important for conservation genomics as a whole.
- Simulations in the original paper detailing the EBSP approach, and those done for our own study, have shown that EBSP performs well with low numbers of individuals as long as the number of loci is large (as in our study).

In addition to these comments, I have some concerns about the relatively small number of species involved. Indeed, although the fact that there is no evidence for a recent decline in the two near-threatened palm species, but there is in the two endangered species, is in line with predictions, the sample size for this analysis is four. Convincingly showing that EBSP analysis can predict conservation status would require a much larger number of species.

We agree that it is too early to say for sure that EBSP can predict conservation status in all instances and we have toned down our statements in the text. We nevertheless have evidence that it does so in our four palms. Our study lays the fundamental groundwork for further research on this theme.

To summarise, this manuscript presents an interesting method that has the potential to change how we assess the conservation status of species/populations. Furthermore, it provides some interesting insights into the demographic history of a number of Madagascan endemics. However, in its current form it is difficult to assess the robustness of the main findings. Furthermore, I am concerned that the manuscript may not be of sufficient interest to the broader readership of Proceedings B: One half of the manuscript is very methodological and hence it be more appropriate for a journal like Molecular Ecology Resources or maybe Methods in Ecology and Evolution. The other half is relevant to the conservation of a specific set of Madagascan species, and thereby more appropriate for a conservation-oriented journal. Having said this, if the method as presented here can be shown to be robust, and there are good reasons to believe that the results obtained for this subset of species are representative of a more general phenomenon, than Proceedings B could be a good fit for this study.

We have provided new simulations for our EBSP analyses (Fig. S2-5) and confidence intervals for our estimates of N_e (Table S3), showing that our results are robust to under/overestimations of N_e , as well as variation in generation time / generation overlap, the main concerns of the reviewers.

The set of species that we sampled come from seven different plant families, to show that our approach is general. Our methodological proposal is widely applicable, so we hope it will lead to use in other regions/taxa.

The multifaceted nature of our manuscript makes it appealing to researchers in different fields, those interested in how genomics can be used in conservation, but also those interested in the history of Madagascar's unique and threatened flora and the origins of the widespread pattern of micro-endemicity. To this end, we have

revealed the possibility of a more widespread pattern in the underestimation of extinction risk than is currently accepted. This has important implications for how we do conservation and the utility of genomic data.

The general interest of the subject can be exemplified by the following related paper (<https://www.pnas.org/content/118/10/e2015096118#sec-6>) that was released while our manuscript was in review. It discusses the importance and usefulness of estimating effective population size and its historical trends for conservation. Our study directly contributes to the ongoing debate about how to best use different aspects of genetic diversity in species conservation.

Reviewer(s)' Comments to Author:

Referee: 1

Comments to the Author(s)

This study addresses the demographical history of Malagasy plants with genomic data, and the question on species rarity over time is a very relevant question in Conservation Biology. Plant endemics are supposedly impacted by an increase of human activities during the last two millennia, and the authors tested population decline during that period. They also suggest that their approach could be used to assess the vulnerability of species for which data are deficient. From genomic data, effective population size (N_e) was estimated and compared to census population size (N_c). A higher N_e than N_c may reflect a rapid, recent population decline. The authors also estimated the demographic history by inferring N_e over time. Their analyses were conducted on several plant models, some with reduced population size and critically endangered, while others seem not be impacted by human activities. Recent population decline signatures were detected in most of species, even in species with low concern such as *Psiadia altissima*. Overall, I consider this work is interesting but based on some strong assumptions that need to be carefully considered.

We thank the reviewer for their helpful comments.

The method has certainly a potential to investigate the recent population demography and vulnerability of micro-endemic species, but I suspect the method could be impacted by many factors that may be discussed, at least briefly.

i) First, the method should be pertinent at the local scale, and thus should not be very useful for assessing the status of plants at the island scale. A comparison of demographic pattern between sites may thus be very informative. The problem is mentioned at line 182, but it should be discussed.

We agree that, in our non-palm species, patterns are at the local scale and have made changes to the text to clarify this (L222). We have also added text about the benefits of comparing among sites (L229).

ii) The habitat of Malagasy plants is generally heterogeneous, and population connectivity could be affected for many reasons, both human-induced and natural

(climate-driven). Many groups of plants belong to species complex, and the native flora of Madagascar is also characterized by many species with limited dispersal capacity. In that context, we could expect that recurrent admixture between taxa may happen after habitat perturbation, or conversely, become suddenly impossible. Such events may have strong impact on the genome, and I'm wondering what could be their consequences on inferences made in the present study. In addition, in plants, and especially woody species, generations can overlap, and may also impact demographic inferences.

These are interesting points that are worth investigating. As such we have added sections detailing the potential effects of population structure (L210). We also agree that overlapping generations can make demographic inference more difficult, though information on the extent to which this occurs in our species is currently lacking. We have added a relevant passage to the text (L165).

iii) Lastly, I supposed that all plant models investigated in this study were diploids. Autopolyploids (or segmental allopolyploids) are quite common and may not be appropriate for such studies.

We investigated the number of alleles within individuals in our ddRADseq data and found in each species 2 was by far the largest proportion. Other congeners of *Dyopsis*, *Ravenea* and *Psychotria* had low c values and ploidy of 2 for the first two genera (Kew c -values database).

Other remarks.

Line 49. Reference "Schatz, 2009" is not numbered and not in the reference list.

This has been fixed.

Line 92. The N_e vs N_c comparison between plants and humans is maybe not so relevant. Tree demography may be impacted by many factors not at play in humans, such as generation overlapping.

We believe that this acts as a useful explanation of the idea for non-experts in the field. Also generation overlapping is quite common in humans [see Mick Jagger]!

Lines 103 and 235. A generation time of 10 years is arbitrary considered. I think the average time for replacing a mature breeding individual could strongly vary among species, but also over time. I also feel this value of 10 years is quite short for the tree species considered in the study.

Many of our species are shrubs so ten years seemed to be an appropriate estimate. For example, ten years was used as a generation time for inference in date palms (Gros-Balthazard et al. 2017) and *Howea* palms (Papadopoulos et al. 2019). Ideally we would have more information about the generation times of our species but this is not the case - though there is evidence that 10 years is a reasonable estimate for small trees/shrubs in the same families as some of our study species (L161). We

have conducted simulations where we have altered the number of generations (see Fig. S2-S5; L169). The lower number of generations before the simulated event would represent an increase in generation time.

Line 205-209. On the choice of species, could you precise their ploidy and the presence of congeners occurring in the study area?

For ploidy of species see above. We did observe congeners in some of our sampling collection sites e.g. for *D. rabepierrei*, but found no evidence for hybridisation.

References

- [8] Wilme => Wilmé (and complete the reference, the journal is not given)
- [14] and [33], complete the references (vol and article no)

This has been fixed

Figures are clear, but a picture is missing for *Psychotria* sp. (Fig. 4).

Unfortunately a picture was not available for this unclassified species. This has been noted in the figure legend.

Line 436. "in in" => "in"

This has been fixed

Referee: 2

Comments to the Author(s)

This paper presents an interesting and clever method to use the effective population size of a species, and its historical change, to obtain information about the extinction risk of species that are otherwise difficult to assess. This is certainly an important issue, as only a minority of species worldwide and across taxa have been assessed, so there are many more at-risk species than we have currently recorded. Overall, the manuscript is easy to follow and flows well. I am pleased to have had the opportunity to review it.

However, there are some shortcomings to the current manuscript. I have some major and minor comments that I hope the authors will find helpful. Please find these attached in a separate document. Most of the minor comments relate to specific line numbers. Some, while linked to specific line numbers, are also of concern more broadly.

Major comments

- It is clear that this method would be useful for currently un-assessed species, particularly those that are challenging to study and get the relevant data for assessment. However it is not clear that this method would add any information, nor re-classify to a more extreme risk category, already listed species.

We agree that it is particularly useful for data-deficient/unassessed species but it is not our goal for the approach to cause a reclassification of listed species by itself. Multiple lines of evidence are important when making conservation assessments and in many cases it is difficult to be sure that the complete picture is observed. It is therefore useful to have another independent source of information on population decline that either supports current assessments or points to potential areas where more work is needed, if genetic data disagrees with the current assessment.

While population size decline is a fundamental part of IUCN categorization, the assessment criteria also include criterion D, “very small or restricted population.” In the previously assessed species in this study (palms), it isn’t clear what this method of calculating effective population size through time adds, since two of the studied species (*Dypsis rabepierrei* and *Satranala decussilvae*) are already classified as Critically Endangered and Endangered, respectively, based on criterion D.

We were using these to check for association between our results and IUCN classifications, as a ‘training’ dataset as we had prior expectations of what our EBSP results should show (this has been clarified: L91).

Classifications can be based on multiple criteria (e.g. B1ab(iii)+2ab(iii); D for *S. decussilvae*), and it is useful to understand why populations are very small or restricted (“have rare species always been rare?”) to improve conservation knowledge and approaches.

A third, *Ravenea robustior* is currently classified as Near Threatened, but according to the stated census size ($N_c=1000$) would now be evaluated as Vulnerable according to Criterion D (population < 1000), while the effective population size method would indicate that it is perhaps a healthy population because its size is increasing.

Again, conservation assessments are not just based on a single criterion, and the reviewer highlights how different aspects of evidence can give different pictures of the status of a species. Our approach adds more information to a species that appears complex. In this study we are not trying to change / rewrite IUCN criteria but instead just add more information to build a more complete picture.

- No Malagasy nor CITES permits are mentioned, nor are the relevant Malagasy ministries acknowledged. This is an oversight which I hope is easily corrected.

The work was conducted via Kew Madagascar Conservation Centre and Royal Botanic Gardens, Kew - a CITES organisation. All work was conducted with the required permits. This has been added to the text (L254).

- While the authors discuss population declines since the settlement of Madagascar as being the main concern, the time scale of the observed declines in the studied species is not discussed. Time scale is challenging with BSPs, since it is strongly influenced by both mutation rate and generation time, and another reader might criticize time estimates for exactly this reason. However, to support the argument that these population declines are anthropogenic in nature, some effort must be made to estimate these timings. Choices of mutation rate and generation time must be strongly justified, perhaps including a range depending on divergent mutation and generation time estimates.

We have strengthened our analyses by adding a range of generation times to our analyses (Fig. S2-S5; Table S3). This has shown that our results are robust to generation time changes, and that when events are very young our ability to infer population size change (as expected) is reduced.

We also note that taking advantage of RADseq in EBSP plots requires that we pick a subset of highly variable RADseq markers. This makes it difficult to assign a mutation rate to infer times. As a solution to this, we simulated data of events in the recent past and showed that our approach can infer declines at this timescale.

Similarly, the authors point out that the resolution of the BSP is limited at very recent time scales, perhaps obscuring recent declines. Unfortunately no effort has been made to determine the limit of that resolution, and thus its importance to these results. This could be remedied with additional simulations – simulate declines of a range of strengths (10%, 50%, 90%, as elsewhere in the paper) starting at a range of times more recent than the current set of simulations, eg. 50, 100, 500, 1000 years bp.

We agree that this is a useful avenue of research and have now added new simulations at 100 and 500 generations to test the limits of our inference. Indeed they were less accurate more recently and more accurate further in the past, as expected (Fig. S2-S5). See L169 for further details.

- The assertion that $N_e \gg N_c$ demonstrates recent population decline must be much better justified, particularly in light of this study's apparently contradictory result for *Ravanea robustior*, which as $N_e \gg N_c$, but the BSP shows no population decline.

We disagree that our results are contradictory but instead complimentary. Given that EBSP is based on the number of coalescent events over time, very recent decline would likely not show up in our EBSP analyses. This is due to lack of resolution caused by the relatively low number of coalescent events in the very recent past. Comparing N_e and N_c does not suffer from this issue and in the case of *R. robustior* or *A. perrieri* may be revealing that decline occurred in the very recent past e.g. <100 generations (L137). By combining these two approaches we can get an idea of trends at a relatively large temporal scale (EBSP) but also an idea of very recent trends (N_e vs N_c).

There are a number of other factors that affect effective population size, and while these factors are typically less important than census/breeding population size in determining effective population size, they can have strong effects under certain circumstances. In addition, effective population size calculated in this manner is extremely dependent on mutation rate. While the mutation rate used in this paper was obtained from a rigorous study of gymnosperm and angiosperm mutation rates, this strong dependence should be acknowledged. Typically, effective population sizes are reported with a 95% confidence range, with the determinants of that range dependent on the N_e calculation used.

We have acknowledged strong dependence on the mutation rate (L112). We have also included a table of 95% confidence intervals for our N_e estimates (Table. S3).

Minor comments

- I am having difficulty finding *Dypsis rabepierrei* on the IUCN redlist. It appears it was newly described in 2018 (Eiserhardt et al. 2018) and has not yet been assessed. If it is listed there under another scientific name, please give that in the text. If it has been assessed but has not yet been added to the online database (next planned update Dec. 10 2020), please give a citation that reflects that. If it is, in fact, as yet unlisted/unassessed, it is inappropriate to make any statements about its classification (even though when it is assessed it will certainly be classified as Critically Endangered due to its census size). If this is the case, please rephrase these statements (eg. Fig. 2b, L111) to reflect this.

We had expected it to be listed some time ago but as this has been delayed we have changed the text as suggested (L97).

- L40-41: Please cite the origin of this data, IUCN, rather than or in addition to the secondary source. In addition, these numbers have substantially changed since the

cited paper (Pimm et al. 2015) was published, there are now 120,372 rather than 71,576 from which those percentages were calculated. See here for updated numbers: https://nc.iucnredlist.org/redlist/content/attachment_files/2020-2_RL_Stats_Table2.pdf

◦ L41: “With fewer than 100k species assessed” - see above. This doesn’t change the point of the sentence, which could even be strengthened by referencing IUCN’s estimates of percentage of described species that have been assessed: https://nc.iucnredlist.org/redlist/content/attachment_files/2020-2_RL_Stats_Table1a.pdf

◦ L43: “Only 27k plant species have IUCN assessments...” see above.

We thank the reviewer for these recommendations and have updated these sections according to the newest red list assessment (2021-1 (L40)).

◦ L45: Schatz 2009 is not in the references. In addition, while the cited statement is very plausible and asserted in the cited paper’s abstract, this reference does not provide evidence or argument for it beyond pointing out that humans are very reliant on agricultural plants.

The citation has been fixed.

The following passage in Schatz 2009 captures what we were aiming for with our citation:

“With the human population optimistically projected to stabilize at between nine and ten billion by the end of the century, food production will need to double at a minimum, and pressure upon wild plants that are already utilized for food, fuel, housing and medicine will correspondingly increase. It is therefore imperative that the conservation status of all wild plants that contribute directly to human livelihoods be assessed and monitored.”

◦ L39-L57: The argument here seems to be that we have not assessed enough plant species relative to their numbers and importance (“Therefore, it is critical that we assess and continue to monitor plant species.”) as justification for using this new method of assessment. It is unclear if you are proposing that using this method would increase the number of plant species assessed (since it would require genomic analyses this seems challenging without concerted effort and funding), or that it would improve current assessments (which does not follow from the argument that we haven’t assessed enough plant species).

We wanted to make two arguments here:

- (1) That we improve the accuracy of current assessments, by making use of genomic data. This would provide comparisons between data types and new perspectives e.g. identifying levels of genomic erosion we were not aware of.

- (2) That we can make novel assessments more efficient by using genomic data as a part of the assessment process (alongside other ways of assessing species risk of extinction)

We have edited the text to reflect this (L54).

◦ L66: There is debate here too, including dates of permanent settlement vs temporary habitation (eg. for hunting). Multiple papers now provide evidence of human presence prior to 2500 years ago (eg. Hansford et al. 2018, 10,500 years bp; Muldoon et al. 2012, 10,000 years bp; interpretation of Goodman et al. 2013 by Godfrey et al. 2019), see Douglass et al. 2019.

We thank the reviewer for pointing this out and have now added the relevant references to our text (L72).

◦ L79: “We investigated population decline...” But some of these species show population increase, not decline, so this is really about trends, not just decline.

We have rephrased this section accordingly (L86).

◦ L94-97: Your skyline plots are a more direct and useful test of population decline than the comparison of effective population size (which in addition to being influenced by breeding population size can also be affected by population structure – location, age, sex; and cyclic population size fluctuations) to census population size. In fact, your BSP result for *Ravenia robustior*, which shows a fairly constant effective population size, suggests that there is something else going on with the effective population size to census size comparison. If $N_e \gg N_c$ is informative of recent decline, shouldn't that be reflected in the BSP for this species?

See our response in the major comments section as to why thetamater results might be different from EBSP in recent history. Additionally, this is an example of why it is good to have multiple tests of whether population size to compare and contrast. Furthermore, *R. robustior* has fewer individuals so estimates of population size are worse and confidence intervals are very large (Fig. 2).

In addition, even assuming these factors are unimportant relative to breeding size in determining N_e in these species, this comparison does not show when this change happened. As you are arguing that anthropogenic population declines are the problem, the timing is very important. In fact, without timing (or additional evidence), one cannot make the statement that these are anthropogenic declines.

See above. We have modified the text to be more pragmatic about whether we can attribute to humans highlighting that the timing is unclear but our results reflect results from simulations of events over anthropogenic timescales (L182).

Not all populations that have declined have done so due to anthropogenic causes, even in Madagascar. Please see Quéméré et al. 2012 for a relevant example.

We have cited Quéméré et al. 2012 as an example of an alternative scenario (L74).

◦ L103-105: You say that “in each of the four palm species”, N_e exceeds N_c , but Figure 2 has the census size of *Dypsis procumbens* as ‘?’ You mentioned *D. procumbens* in L87-88, but there is no census size there either. Lack of census size is understandable, but please be more precise in text.

We have clarified this (L115).

◦ L103: Generation time is very important here, please cite or otherwise justify choice of 10 years.

Many of our species are shrubs so ten years seemed to be an appropriate estimate. For example, ten years was used as a generation time for inference in date palms (Gros-Balthazard et al. 2017) and *Howea* palms (Papadopulos et al. 2019). Ideally we would have more information about the generation times of our species but this is not the case - though there is evidence that 10 years is a reasonable estimate for small trees/shrubs in the same families as some of our study species (L164). We have conducted simulations where we have altered the number of generations (see Fig. S2-S5). The lower number of generations before the simulated event would represent an increase in generation time.

We have also now tested the effect of increased generation times on the N_e estimates (Table S3; L167) and the patterns in our results did not change. Generation times would have to be extremely large to change the $N_e > N_c$ pattern observed so we think that it is unlikely to have an effect on this section of the paper.

◦ L104: table S1 reports neither effective population sizes nor census sizes. Did you mean a different table? (I don't see N_e in the other supplementary tables either.)

We have now included a table with census sizes and estimated N_e with confidence intervals (Table S3).

◦ L111-112: “Given that our N_e estimates and IUCN assessment suggest our four palm species have rapidly decreasing populations...” IUCN states that the current population trend for *Satranala decussilvae* is not known – the classification as Endangered is based on criteria 2 (agricultural habitat loss) and 5 (resource use), not population decline (which, while likely due habitat decline, is not known). I am unable to find a citation supporting the population decline statement for *Dypsis rabepierrei* either.

The statement related to IUCN classification was incorrect as the reviewer pointed out and has been removed.

◦ L122, L124: Table S2 does not have this information.

Tables have been reordered/updated.

◦ L125: What is the time limit of your EBSP resolution? That is, how recent would the decline have to be, to be undetectable? Your simulations may already be able to provide this. While it is certainly plausible, I would like to see more discussion about the current threats to these species, for example whether their habitat has recently seen a strong decline or fragmentation, within that time range of N_e change undetectability. If they have, this would support your statement that this could be an undetected decline. However, your simulations suggest that EBSP is more likely to detect size changes in these two species (*Dypsis procumbens*, *Ravenea robustior*), with loci of 380bp and 598bp, than in some of your other species, eg. *Satranala decussilvae* (147bp).

Our simulations suggest that for events more recent than 100 generations patterns become more difficult to detect, depending on read length (Fig S2-5). We have detailed the threats to the species and have suggested that recent evidence put forward by the IUCN could explain undetected decline (e.g. L199).

◦ L127-128: I am not sure what “more extreme levels of risk” means in this context. These two species are already listed as Endangered and Critically Endangered. Do you perhaps mean that these results suggest that this method would give us the ability to assess the extinction risk of currently unlisted species?

This isn't what we meant, we have deleted the word “more” and clarified (L141).

◦ L151-172: This is very straightforward and easy for me to follow. This seems like the most appropriate application for your method – assessing currently unlisted species, or adding to assessments of data deficient species. However, the percent declines mentioned here (and in figures 2 and 4) are somewhat misleading as currently calculated – these are described as the percentage decline from maximum population size to minimum inferred after the maximum, but for about half of the studied species (particularly those that show recent declines), the minimum inferred population sizes also have extremely large HPD intervals, making median estimates, and thus percentage decline calculations, meaningless.

We disagree that it is meaningless, but agree that considering HPD intervals are important. HPD intervals are expected to increase very close to the present due to a lack of very recent coalescent events. By comparing N_e and N_c we are able to provide some perspective during this difficult to estimate period, which would suggest that declining species with large HPD intervals continued to decline, even when we lack signal in the EBSP.

We have added to the text explicitly stating that these are median estimates, and that HPD intervals were wide in some species so some caution should be used when interpreting decline proportions (L179).

◦ L223: You mention locus lengths of 147bp or 380bp, but Table S2 reports *Ravenea robustior* with a locus length of 598bp.

We have fixed this.

- L239: If only loci with high numbers of SNPs were used, this may have biased toward longer loci (L222-223, “Read pairs were concatenated to produce a single sequence of either 147bp or 380bp...”)

All loci belonging to a species were the same length, but length of sequenced loci did vary among species. We realise our statement was unclear and have rewritten it to clarify what was meant (L271).

- L254-262: Some of the samples come from locations distant from each other (*Dyopsis procumbens*, *Ravenea robustior*, *Satranala decussilvae*), and sampling schemes that pool samples from genetically divergent populations are known to sometimes cause BSP steps that can be spuriously interpreted as population size change (see Grant 2015 for explanation). The described simulations do not seem to take this pooling into account for these species.

It is indeed possible that population structure may influence demographic inference, and this should have been mentioned. We now cover and discuss potential effects in the text (L210) and note that our N_e vs N_c analyses support inferences of decline in EBSPs. It would be interesting to simulate how sampling diverse populations would affect demographic inference but this is out of the scope of our study.

In regard to Grant 2015, this paper is interesting but is aimed at BSP (rather than EBSP), with a focus on plastid/mitochondrial data, rather than the relatively large number of nuclear markers used here.

- L417-418: Fig. 2 “Results of skyline plots for simulated data are shown as coloured” Coloured what? The colored bands around the estimates seem to be the 95% HPD, so I am not sure where the simulated results are in this figure.

This was a typo and has been fixed

- Fig. S2: This figure could benefit from including the results of the original runs (just the median values, not the entire HPD), perhaps as a dashed line.

We have added the median values as suggested.

Referee: 3

Comments to the Author(s)

In the present study, ddRAD data on 43 individuals representing 10 species are analyzed to estimate present and past effective population size. Furthermore, simulations are run to investigate, if with the available dataset, changes in population size can be detected.

Where census is available, the authors generally find much higher estimates of effective population size than census size. They interpret this observation as evidence for recent drastic population declines. This would suggest that many currently rare endemic species in Madagascar were potentially much more abundant in the past.

This is per se an interesting study once more pointing out the importance to not only consider census but also genetic information when assessing a species extinction risk and demonstrating the potential of low-budget genomic projects in doing so. However, I am not entirely convinced by the analysis. I regard the information currently provided as not sufficient to support the conclusions and I am not sure if the sampling scheme of only very few individuals per species is enough.

One of the main goals of our study was to test the limits of working with rare species. We did this empirically and by conducting simulations with low numbers of individuals. We have now added new simulations assessing the effect of when declines look place and have shown with that it is possible to detect signals of decline using our approach (Fig. 3, S2-5).

We understand the reviewer's concern as in an ideal scenario (with model organisms for example) we would infer patterns with many more individuals. However, assessing what we can infer with a paucity of samples is an important avenue of research when dealing with rare species, those for which there is limited data but often those we want to conserve the most.

Furthermore, a more critical discussion of the EBSP analyses (in particular regarding effects of gene flow and population structure) is needed.

We have added a discussion of the potential effects of population structure on EBSP analyses (L211). We also point out that one advantage of working with species that are rare or from a small geographic range is that the effect of recent gene flow and structure is minimized.

My main concerns are the following:

Ne estimates in general, and Bayesian skyline plots in particular, are sensitive to population structure and gene flow. Locally estimating a larger Ne than the observed census size may also be indicative of gene flow with another population or population structure among sampling sites.

Also the shape of skyline plots can be highly influenced by these factors (see for instance Heller et al. 2013). At least some discussion of these caveats is important to add.

See our response just above. The census size available is for the whole species, rather than just the local population. We have made this point clearer in the text (L116). We have now added text discussing the potential effect of population structure on our inferences (L211) and suggest that inferring the level of gene flow and its potential effect is a useful avenue of future research (L217).

In the discussion (l. 178-181) the authors admit, that some of these population size estimates may not really be representative of the entire species. This seems a bit in contradiction with the general conclusion about past abundance of current rare endemic species.

For rare species like *S. decussilvae* or *D. rabepierrei* we have sampled the majority (perhaps the entirety) of their very small range. We agree with the reviewer that in some species where sampling does not cover a large proportion of the species' range, we may be only inferring local patterns. This will give us a good estimate of what the species is doing in the absence of extensive sampling (often the case in rare species) and is therefore still critical information for conservation of species' populations. We have added some additional discussion to the scale at which our inferences apply and highlighted the potential for future work in this area (L225).

I find the results presented in the main figures (2-4) hard to interpret. The units are not comparable with the N_e and N_c estimates given above. I quickly tried to estimate N_e for figure 2a using the y-axis and the mutation rate given in the methods, but this resulted in N_e values close to 2 mio. I am surely doing something wrong... Please give useful units (plain N_e) and generations or years for time. Also, what explains the general increase of N_e over the time frame analysed?

EBSP methods produce a value of effective population size scaled by mutation rate, which makes conversions more complex. The units provided are standard for uncalibrated EBSP analyses. Furthermore, taking advantage of RAD sequencing in EBSP plots requires that we pick a subset of highly variable RADseq markers. This makes it difficult to assign a mutation rate to infer accurate timescales. As a solution to this, we have added a range of simulations of events in the recent past (100-500 generations) and demonstrate that the technique can infer declines at this timescale with inference becoming more difficult, as expected, in the recent past (L174).

The general increase towards the right hand side of the graphs is the result of the species/population origin as well as the limit of inferred coalescent events as they reduce in frequency far in the past (hence the high uncertainty in estimates). This pattern also holds true for the very recent past when coalescent events are also rare.

Furthermore, I think clarification is needed regarding the raw data analysis. For instance, what was the coverage at the loci used for the analyses? Coverage can have important impacts on diversity and hence N_e estimates.

We have added information about average coverage to Table S2. Our coverage was generally high, and >25x in all species, so we can be confident in our sequencing.

It didn't come clear to me, what is meant by the different "read lengths" (147bp, 380bp, 598bp), how they got obtained (see also my comment for l. 223). Where does the difference among species come from (different sequencing efforts?)?. I also think there could be a correlation of this measure with diversity measure (He, etc), please check.

Differences in read lengths were due to the limitations of different sequencing kits, rather than a specific methodological choice. We compared genetic diversity statistics to read length (found in Table S2) and found no evidence for a correlation.

The raw data analysis was performed per species and therefore each time with only very few samples (2 to 5 samples). I expect that disentangling sequencing errors from real SNPs is very difficult with so few samples while sequencing errors are problematic for Bayesian skyline plots.

At low coverage this would be a problem but given our high coverage (see above) we do not think this is an issue as bases are accurately called. Furthermore we are using up to 50 RAD tags per analysis so any sequencing errors that slip through will be drowned out by the rest of the data - low frequency variants from error wouldn't have an effect on general trends.

The possibility of simply underestimating census size, because some may be difficult to find, should be discussed a bit more.

We agree that this is an important point, and have now made it in the text (L117).

Specific comments:

I think it would be helpful to have some of the information given in Table S2 in the main text (Species name, sample number and diversity measurements).

We have added sample numbers to figures 2 and 4. We would prefer to keep the table in the supplement as now much of this information is in the main text / figures.

l. 23: some sort of break needed between extinctions and yet?

This has been fixed.

l. 26: as Ne estimates can be tricky, I suggest to mention in the abstract, that this study used ddRAD.

We now mention we used RADseq in the abstract.

l. 28-29: or over/underestimation of one of the two...

We now present 95% confidence intervals of our N_e estimates (Table S3), which suggests it would be unlikely that under/overestimation is a major issue in our case.

I. 32: does this refer to simulated data? and declines of what?

This has now been clarified in the abstract

I. 81 decreased in area?

Yes, this has been clarified.

I. 85: what is meant by training data set?

This has now been explained (L92).

I. 103: as the generation time is important here, some information is needed how it was chosen and how accurate it may be to use the same for all these plants.

We have added text explaining how generation time was chosen and its accuracy (L160). We have also added additional estimates of N_e with increased generation time (see Table S3) that show our results are robust to changes in N_e .

I. 105: Could this also be explained by gene flow among populations/species? Also, a naive question from a non-botanist: how accurate is census size? From the pictures on Figure 2 it seems well possible, that these plants are not very easy to find? To also give census over time would be informative.

In the rare/endangered Palms gene flow is unlikely to play a role as they are found in a single location. Gene flow could increase diversity in some other cases but given the large disparity between N_e and N_c it is very unlikely to have solely driven the observed patterns. Investigating the amount of gene flow among isolated populations would be an interesting avenue of further research.

Census sizes are taken from the IUCN red list and are our current best estimates. Given that these are provided by world experts in these species we have confidence in their accuracy. Census over time data would be indeed interesting but is currently not available for the species we use.

I. 118: Please explain, what is meant by the "sum(indicstors.alltrees)".

This is explained (L150).

.

I. 125: which endangered species do the authors mean? endangered palm trees or other plants? And are these declines in N_c or N_e ? Is there a figure or table showing this?

We have clarified we meant endangered palms, that Ne was referred to and referenced figure 2.

I. 164-166: but even in the simulations with 10% increase, there was a small decline?

This is due to the expected lack of coalescent events near the present.

I. 200-201: this is contradicted by I. 178-181, where the authors admit that some of these population estimates may not represent the entire species.

This is not a contradiction as these are our best estimates of species-level processes, but we have expanded the point that they may only represent local-scale processes (L225) if other populations (if they exist) are doing different things. Furthermore, this is unlikely to be the case in palms where we have widespread sampling (Fig. 1).

I. 205: seven plant families?

We did test seven plant families but at the end we are proposing that the pattern is more widespread (to investigate in future work).

I. 223: I assume, the 380bp is without the adapters, but 147bp would be much smaller than expected from the library size (~500bp)? Also, I see how overlapping reads can be concatenated (or merged), but to merge the 75bp, the introduction of Ns would have been necessary with this library size?

This was unclear, and we have now added a section detailing for each species their read lengths during sequencing and the concatenated marker size used (L269; Table S2).

I. 224: Aligned to the concatenated tags? I read between lines that this was a denovo analysis, ie. no reference genomes used? If yes, please specify for people who, as me, never used Stacks. Could you give somewhere a range of approximate genome sizes?

We have now specified our analysis was de novo (L272). However, no information about genome sizes is available for any of our species of interest.

I. 224: How well does Stacks perform with such low sample sizes per species? I would assume that SNP calling (disentangling SNPs from sequencing errors) is tricky with few samples. Sequencing errors are, to my knowledge problematic for Bayesian skyline plots.

Given our high sequencing coverage errors in SNP calling should not be a major issue (see above).

I. 230: ...output of from the...

This has been fixed.

I. 230: how were diploids (or even polyploids?) handled? Was there one or more fasta sequences per individual?

One haplotype was chosen at random for each individual at heterozygous loci to minimize bias in haplotype frequencies and reduce bias caused by undiscovered heterozygote samples (L294), as recommended in Trucchi et al. 2014.

I. 239: I see that the authors cite another paper for this so my question is probably naive. Still, by focusing on regions with high diversity, isn't there the risk of overestimating global diversity and hence N_e ?

This is not at all naive and we should have better distinguished between the different estimates of N_e . The former analyses used the entire RAD dataset for each species, which is what we compared to census size. EBSP used the subset of highly variable loci (not compared to census size). We have stressed this point in the text (L284; L293).

I. 241: why was only a very small subset of the loci chosen?

This is due to limitations of EBSP approach (computationally too intensive after ~50 loci) and, to a lesser extent, the number of highly variable loci available per species. This has been highlighted in the text (L293).

I. 244: what is meant by SNP class?

This should have read SNP category, which is now explained (L296).

I. 247: were the replicate runs performed with different sets of 50 loci (I. 241)?

Yes

I. 250: does this mean that for most species, only 50 loci were available?

The number of highly variable loci was lower than the 100 required for two analyses in 4 of 10 species (those not included in Fig. S6). We have clarified this (L303).

I. 255: is the generation time for all these different species really always about 10 or were the authors thinking about a particular species?

Many of our species are shrubs so ten years seemed to be an appropriate estimate. For example, ten years was used as a generation time for inference in date palms (Gros-Balthazard et al. 2017) and *Howea* palms (Papadopoulos et al. 2019). Ideally we would have more information about the generation times of our species but this is not the case - though there is evidence that 10 years is a reasonable estimate for small trees/shrubs in the same families as some of our study species (L164). We have conducted simulations where we have altered the number of generations (see

Fig. S2-S5; L169). The lower number of generations before the simulated event would represent an increase in generation time.

We have also now tested the effect of increased generation times on the maternal N_e estimates (Table S3; L177) and the patterns in our results did not change. Generation times would have to be extremely large to change the $N_e > N_c$ pattern observed so we think that it is unlikely to have an effect on this section of the paper.

I. 257: ...estimates for? *D. procumbens*...

This has been fixed

I. 259: what is meant by SNP category here?

This is now explained (L296).

Figure 2, y axis and IUCN status are difficult to read.
What do the bp numbers mean?

We have increased text size where possible (in Figure 4 as well).
We have explained that numbers shown are marker size for each species' dataset.

I. 417/418: where can this be found?
Could the time be shown in generations or years?

Axes are now explained in the legend. Due to the highly variable loci chosen it is not possible to convert times into generations / years (see above).

Figure 3: what do the different lines per panel represent? The legend in d) represents the four different panels I assume? It would be very interesting to see, what scenario led to an increase and then decrease of pop size. It may be more intuitive to compare this figure with the empirical data, if also here a logarithmic scale was shown. Or is then no change visible anymore?

We have detailed in the legend that different lines represent different simulation runs, and explained the legend in panel (d).

We have changed the scale to log scale for our simulation plots (Fig 3, S2-5).

Appendix B

Associate Editor

Comments to Author:

Many thanks for submitting a revised version of your manuscript. As you will see below, your revision has been reviewed by two reviewers, both of which also reviewed the original submission.

Like both reviewers, I really appreciate the effort that has gone into this revision. In particular the power analyses are an immensely valuable addition, and we all agree that these go a long way in addressing concerns regarding the robustness of your results. Nevertheless, you will find that both reviewers remain relatively critical of some of your results and their interpretation.

We thank the editor for their constructive feedback.

For example, Reviewer 1 argues that the recent declines in population size in may be overestimates, and that population structure may have inflated estimates of N_e . Although this is briefly discussed in the revision, it would be worth expanding this section and/or making it more prominent (also see my comment regarding the structure below).

We have expanded upon how structure might affect estimates of \$N_e\$ by forming a new subsection of the results and discussion part of the manuscript that directly addresses these caveats (L299).

Reviewer 1 also still misses some important information on how the raw data was analysed, and the filtering in particular.

We provided additional details on how the raw data was analysed and filtered (L119-130). Generally, we took a minimalistic approach to filtering in order to maximize the amount of data available and avoid introducing artifacts caused by the limited number of individuals per species.

Reviewer 2 also remains rather critical and raises a number of further concerns regarding the interpretation of the simulation results and the high uncertainty around some of the estimates.

We have included new passages of text to address this uncertainty stemming from difficulties inferring, and simulating, population size changes in the recent past (L235; L261).

Furthermore, like Reviewer 1, they remain concerned about the potential effect of population structure.

As mentioned above we have focused more on the potential impact of population structure on inferences in the new subsection (L299).

Although several of the issues raised are pretty substantial, I also believe that many of these could be addressed in a further revision that either removes the concerns

raised, or directly acknowledges them. Almost all studies will have some caveats and/or rely on potentially oversimplistic assumptions, but as long as they are clearly acknowledged and it is made clear what can be learned from this study despite its limitations, they are a valuable contribution to the literature.

As suggested, we have incorporated caveats in the appropriate places and acknowledgement of limitations requested by the reviewers (see below). We hope that it is now clear what can and cannot be established with these kinds of datasets/analyses, and how this information can be useful despite the limitations.

This brings me to my concern that your manuscript may be too methodological/specialist for a journal like *Proceeding B*. Indeed, the addition of the simulations and the expanded discussion of the robustness of the results has resulted in a manuscript that is arguably more technical and method-oriented than the previous version. Having said this, I also appreciate the open and transparent discussion of what we can and cannot learn from analyses such as these. Furthermore, I found that the introduction makes a convincing argument for why we need alternative methods to assess population trends and to infer extinction risk, and for why genomic methods may provide such an alternative. In fact, your significance and general interest statement in the cover are very strong and well-written, and I think it would be a pity not to incorporate them into the manuscript itself.

We have now incorporated our significance and general interest statement into the main text (primarily the introduction and conclusions sections) to accentuate how our study is of general interest.

This brings me to the current structure of your manuscript. Your decision to combine Results and Discussion and to present these before the Methods is to some degree a matter of taste, but given its large methodological component, would it not be more appropriate to make the Methods section more prominent by moving it forward, instead of 'hiding' it at the end of the manuscript?

We have reformatted the manuscript to put the methods in front of the results & discussion.

Indeed, I think there still is some room for improvement when it comes to the structure of the manuscript in general: The Introduction currently consists of only two, very long and rather dense paragraphs. Would you be able to split these into multiple shorter paragraphs? The same is true for several paragraphs in the Results and Discussion section. I think this should be pretty straightforward and it would significantly improve readability. I also think that several sections in the Results and Discussion would be more at home in the Introduction, such as the general explanation of what we can learn from the comparison of effective and census population size and previous examples of this approach.

Currently the Introduction is one manuscript page, and the Results and Discussion covers five pages that include everything from methodological details to general conclusions. I suspect that adopting a more 'traditional' structure, shorter paragraphs

and maybe even some subheadings, will make your manuscript accessible to a wider readership: It will allow people to read only those sections relevant to their interests, and at the same time important details are less likely to get snowed under.

In line with these suggestions we have reformatted the manuscript to a more traditional structure. As well as moving the methods section we have split some of the longer paragraphs. We have also added subheadings in the results and discussion to make it easier to follow and transferred some sections from the results/discussion to the introduction, as recommended (e.g. L58-62 ; L86-92).

Finally, although the mention of permits etc. for collection and export is an important addition, it currently is extremely generic. I think this should really be accompanied by some form of proof, e.g. by listing permit numbers and providing details on who provided the permits. You give more details in the response to the reviewers, but this (and ideally more) should be included in the manuscript itself.

We have now added the relevant research and CITES permit numbers to table S1.

Also, Figure 1 could be improved: The information content is very low and I wonder if it is needed as part of the main text at all. If it remains, is there an alternative to colour-coding the species? Different symbols or letters? Arrows?

Although there is limited information in the figure, we think it is useful to have an overview of where sampling took place, particularly given the focus on the potential effect of population structure. To improve the figure, we have modified Fig 1 to show distribution of rain forests, added a picture of the main study area and used symbols to differentiate species more easily.

Reviewer(s)' Comments to Author:

Referee: 3

Comments to the Author(s).

The authors have responded to most of my comments and the manuscript has improved.

We thank the reviewer for their constructive and useful comments.

However, I remain with some rather major comments.

I now see why my trial to estimate N_e from the EBSF results didn't work. I was using the wrong mutation rate as these estimates are based on highly variable RAD tags. When comparing absolute values of N_e and N_c , the mutation rate is a crucial player. The authors just explain, why they didn't have a better estimate, but they definitely need to show, how robust their main results are to different mutation rates and properly discuss this issue.

We have reformatted our discussion to explicitly refer to, and expand on, the potential effects of mutation rate and generation time in a subsection of the results and discussion (L239). We note that changing the generation time is equivalent to changing to the mutation rate when using theta as the summary statistic. So, although we are not changing mutation rate explicitly in our simulations or our thetamater estimates of \$N_e\$, our methods still allow us to say something about its effect.

I was surprised to read in the response to me that gene flow and population structure are not relevant, if at the same time, the authors did add it to the discussion as response to reviewer 1. I am glad they added some discussion on this.

What I am missing a bit, is that a few clustered populations (hence population structure) could lead to an overestimation of N_e , also in the case of the direct estimation using thetamater.

We reformatted our results and discussion to have a section on the potential effect of structure to make our points clearer (L299).

Given that most of our species are taken from a single population, the only species where there are a few clustered populations used would be *D. procumbens* and *R. robustior* (though given that only two individuals are used the estimates of \$N_e\$ are already relatively uncertain). As with generation time and mutation rate, to change the general pattern of \$N_e > N_c\$, structure (alongside mutation rate and generation time) would need to be inflating \$N_e\$ by 880% (L248), which seems unlikely.

Also, declines in coalescence-based N_e estimates in the recent past is often observed and can simply be due to local effects. Ancestors from the close past are more likely to be from nearby and hence more likely to be genetically similar.

We acknowledge that by sampling in restricted areas our results may not be representative of whole species but local populations (L314). Though if local effects were indeed driving the patterns observed, we would expect to recover similar patterns in all of our species with local sampling, but instead have a range of demographic histories (Fig. 3) and variable N_e estimates.

The authors have improved the presentation of the raw data analysis. An information, which would help readers, who don't know stacks, is if it also retains monomorphic loci. I believe this is important because of the mutation rate used for the thetamateR N_e estimates. I assume the mutation rate used is a genome-wide estimate, so including invariants sites would be correct?

We have added details to the methods section (e.g. parameter settings un ustacks, monomorphic loci are included, options -r 1 and -p 1 used to filter populations output) (L119-126).

Concerning the raw data analysis, I am still missing what kind of filtering was performed. Were singletons kept? Was there any filtering for heterozygosity? De novo analysis of RAD data has the risk to lead to paralogous sites. This could inflate N_e estimates.

We performed little filtering on RAD loci. We wanted to use the maximum amount of genomic data given that we were working with a small number of individuals.

Related: I still find the use of only highly variable RAD tags to infer past N_e a little scary. 3 or more SNPs per locus seems like a lot and the risk to include paralogous sequences high. Please give the heterozygosity distribution at RAD tags used for the N_e estimates and for the EBSP analyses.

We believe the risk of a large effect of paralogous sites on diversity measures is relatively low in our dataset, for several reasons :

- We used $m = 5$, $M = 4$ in ustacks (now noted L121). Verdu et al. 2016 suggested that for “M” a value of 4 “loci are neither too oversplit nor too merged.” Increased values of “m” increased the possibility of merging paralogs. Given that our m is relatively low (5) it likely lowers the potential for misinterpreting paralogs, as suggested in Verdu et al.
- We enabled the deleveraging algorithm in ustacks, used for resolving over merged tags (now noted L123).
- We found little evidence for polyploidy in our species. In a study that looked at paralogs in other non-model crab species, putative paralogs make up $< 3\%$ ($330/12435 = 0.027$; Ravindran et al. 2018) of the dataset. The potential for paralogs to affect inference is much higher in species where there has been an ancestral genome duplication e.g. Atlantic salmon (Ravindran et al. 2018).
- H_e and H_o are not particularly high (see table S2) and values of H_e and H_o are similar. As per your suggestion we compared the H_e distribution of all loci vs EBSP loci only and found that they were generally similar between the two

datasets. We have added a figure to the supplement showing these distributions (Figures S1).

- We use at least 50 loci (sometimes twice this) for our EBSP analyses, and all available loci for Ne estimates, so if we did have any paralogous loci merging we would hope the signal of the true orthologs would drown out any noise from paralogs.

In summary, the stacks pipeline prevents over-merging but indeed even the most accurate approaches cannot remove all potential paralogy (Ravindran et al. 2018) so this is a potential source of error/limitation of data. We have noted that do novo analysis of RAD data could possibly lead to inflated diversity estimates through merging paralogous sites (L122).

In the simulations, the expansion is not detected. The authors explain this to me by lack of coalescence events in the past. This means that their method can only show declines, even in a scenario of expansion. This should at least be mentioned somewhere in the text.

We have now mentioned difficulties in inferring expansions in the recent past in the text (L235).

Related to this: is the number of replicates displayed in Figure 3a the same as in b-d?

We have specified the number of replicates for each panel in Figure 3a (as well as for the figures S2-6.)

And rather a comment to the author response regarding sample size: Of course low sample size is a common issue in conservation. To prove a principle, it could then be better to do such analysis using a more abundant species and infer population size by down-/resampling.

Indeed, we plan to do this in the future, but more abundant species will likely have very different demographic histories than rare species, even when subset, and so will be less useful for uncovering how rare species become rare.

I. 152-154: Wouldn't it be correct to reject constant size if there was a change in population size, even if it wasn't severe?

This is possible, but we chose to take a more conservative approach based on the "sum(indicators.alltrees)" statistic to avoid inferring spurious changes in population size.

I. 314: with pipeline above, the entire pipeline from SNP calling to Ne estimates is meant?

We have clarified that we meant the EBSP pipeline, replacing the output from populations.

Table S2: Please indicate number of variable sites as well

We have added the number of variable sites.

Referee: 2

Comments to the Author(s).

I appreciate the authors' efforts to respond to all three reviewers, especially given the detailed comments. In addition, the authors provide additional simulations to show that mis-specification of the generation time does not change the comparisons between N_e and N_c , and provide discussion about both the effects of mutation rate and population structure. They show that their sample size is adequate for the EBS-P analyses they perform, and provide confidence intervals for the N_e s they calculate.

We thank the reviewer for their insightful comments.

However, I have some remaining concerns, and some new concerns brought up by the additional information provided in response to my and the other reviewer's comments.

First, the authors clarify in their response that this work is not intended to "change/rewrite" IUCN assessment criteria, but "just add more information" for current and novel assessments. While more information is always useful, some concrete examples of how this additional information might be used would be beneficial here to show the general use of this method. As the authors point out in their response, there are indeed multiple criteria and subcriteria for assessment, simply adding another piece of information does not seem of broad enough interest.

We have outlined scenarios where the additional information provided by our approach will be useful (L333-340). This primarily comes down to the efficacy of our approach, there are too many species and not enough specialists for each endangered species to be given the amount of time, money and effort for an IUCN assessment. For example, one way our approach might be useful is to identify species that are in urgent need of more thorough assessment.

Furthermore, we have no understanding about what happened to most species before 100 years ago (or before records began). Identifying whether the rarity of a species is implicit in their biology can also help to better direct conservation efforts towards species that experienced recent population decline.

One possibility might be for direct conservation management. A species that has recently become rare might benefit from a breeding and reintroduction program to increase its numbers, while a microendemic species that has always been rare would not benefit from this, and efforts could more usefully be put toward protecting its existing habitat and individuals. This would have the potential to change how conservation management is conducted, with substantial and fast results.

We agree and thank the reviewer for this useful suggestion and have added it to the manuscript (L334).

I very much appreciate the additional simulations the authors performed, since these can be quite time consuming. Some of these simulations show that a population

expansion of 10% can give an EBS that looks similar to decline of about the same magnitude, and the estimates of the magnitude of decline are vary variable (eg. -0.21 to -0.80 for a 10% decline with 380 bp reads, -0.53 to -0.96 for a 10% decline with 147 bp reads) so I now wonder how reliable the EBS conclusions of population declines and there magnitudes are. For example, the decline estimated in *Satranala decussilvae*, concluded to have undergone a 78% population decline, falls within the range of declines simulated for simulations of 10% expansion, 10% decline, and 50% decline.

We have now noted that at smaller read lengths and in more recent history even expansions can be inferred as declines (L235-238).

Fortunately, such reconstructions are not the only information to take account of. Other sources (N_c vs N_e , biological knowledge e.g. IUCN) can help us get a better idea if trends are reliable, particularly in the case of *S. decussilvae*, where we know a great deal about their biology and recent history, so can be more confident in the declines inferred.

Regarding the additional information provided in Supplemental Table 3, the 95% confidence limits of the estimates of theta include 0 in 7 of the 10 species, but the listed N_e estimates do not. The formula for calculating N_e from Θ is $N_e = \Theta/4\mu$, so when $\Theta = 0$, $N_e = 0$ as well, unless there are typos for these seven estimates of the lower CI or some additional calculations are going on other than those described in the ThetaMater documentation. If the true CIs do include 0 (or indeed the N_c of that species), then reported differences between N_e and N_c are not significant.

This was a typo as the values were truncated and has now been fixed. The true CIs do not include 0 or the N_c of the species, so the significance remains the same.

Lastly, while the authors do comment on how population structure could affect these patterns, I find their discussion to be insufficient to the potential for confounding. They cite Heller et al. 2013, who found that in structured populations, EBSs with only local samples of a population, as well as pooled samples, can exhibit false signals of population decline (Fig. 1 A-E), though the effect in pooled samples is more pronounced with very low migration, and scattered samples do not. They define local samples as all samples from a single deme, pooled samples as multiple samples from a subset of demes, and scattered samples as one sample from each deme in the population. The authors comment that their locally-sampled species may therefore be subject to these false signals, but not the palms which were sampled from multiple demes. Unless I am misunderstanding the authors' sampling scheme in this work, the sampling scheme of the palms may more closely match Heller et al's definition of "pooled" samples, especially for *Satranala decussilvae* (2 samples each from 2 populations).

We agree with the reviewer that 'pooled sampling' is probably a better representation of our palm sampling. We have revised this passage to better reflect the potential caveats behind our pooled sampling of palms (L306).

Furthermore, Mazet et al. 2016 (Heredity) show that “any demographic model with structure will necessarily be interpreted as a series of changes in population size by inference methods ignoring structure.” While I agree with the authors that investigating gene flow among populations and determining if and how it would affect estimates of demographic change and N_e would be an interesting avenue for a comprehensive future study, I maintain that some analytical treatment is necessary to show the general utility of this method. This is especially important since populations of conservation concern are frequently restricted to habitat fragments, and therefore may be subject to strong, anthropogenic population structure, in a time frame comparable to the time frame of the declines being investigated.

We agree that getting a better understanding of how population structure affects demographic estimates in rare species is useful, as we highlight (L321), but this is simply not possible with our current sampling. Furthermore, in species with very restricted ranges, known from a single population e.g. *D. rabepierrei*, this may never be possible. We have suggested that incorporating genetic structure would further the applicability of our methods (L325).

From a fundamental evolutionary perspective the effect of structure is interesting and important. From a conservation perspective a species that is subject to anthropogenic population structure is also key to identify and still helps target conservation efforts. If a rare species is also highly structured it would still be in need of substantial conservation effort, so ultimately the results will work towards the same goal, even if the relative effect of different phenomena on estimated effective population size decline remain unclear. Teasing apart the relative contribution of these different factors in rare species is another useful avenue of research, which we aim to explore after the reviewers' feedback.